# Reliable Probabilistic Forecasting of Irregular Time Series through Marginalization-Consistent Flows

**Vijaya Krishna Yalavarthi**[* 1,2], **Randolf Scholz**[* 2],
**Christian Klötergens** [1,2], **Kiran Madhusudhanan** [1,2], **Lars Schmidt-Thieme** [1,2]
[1]Information Systems and Machine Learning Lab (ISMLL), University of Hildesheim, Germany
[2]VWFS - Data Analysis & Research Center (DARC), University of Hildesheim, Germany
`{yalavarthi, rscholz, kloetergens, madhusudhanan, schmidt-thieme}@ismll.de`

**Stefan Born**
Institute of Mathematics, TU Berlin, Germany
`born@math.tu-berlin.de`

## Abstract

Probabilistic forecasting of joint distributions for irregular time series with missing values is an underexplored area in machine learning. Existing models, such as Gaussian Process Regression and ProFITi, are limited: while ProFITi is highly expressive due to its use of normalizing flows, it often produces unrealistic predictions because it lacks marginalization consistency—marginal distributions of subsets of variables may not match those predicted directly, leading to inaccurate marginal forecasts when trained on joints. We propose MOSES (Mixtures of Separable Flows), a novel model that parametrizes a stochastic process via a mixture of normalizing flows, where each component combines a latent multivariate Gaussian with separable univariate transformations. This design allows MOSES to be analytically marginalized, enabling accurate and reliable predictions for various probabilistic queries. Thanks to its inherent marginalization consistency, MOSES significantly outperforms all baselines—including ProFITi—on marginal predictions. For joint predictions, it beats all other consistent models and performs close to or slightly worse than ProFITi. Implementation details: `https://github.com/yalavarthivk/separable_flows`

## 1 Introduction

Probabilistic forecasting of irregular time series requires models that can make predictions at *arbitrary* time points and for an arbitrary subset of variables (channels). Unlike regular time series forecasting, where predictions are confined to a fixed grid, irregular settings require modeling a **stochastic process** as the number of targets can vary.

A core requirement of any stochastic process model is **marginalization consistency**: the marginal distribution of a subset of variables must agree whether computed directly or by integrating out the joint. This property, a precondition for **Kolmogorov's Extension Theorem**, ensures that the model defines a mathematically coherent stochastic process. Consistency is not only necessary for mathematical rigor but also highly desirable in practice: by the **data processing inequality**, accurate joint predictions together with consistency ensure reliable marginals, providing both correctness and performance guarantees.

Many recent methods prioritize flexibility over consistency. For example, ProFITi (Yalavarthi et al., 2025) achieves strong empirical performance using normalizing flows but violates marginalization consistency, producing contradictory marginal and joint predictions (Figure 1).

---

[*]Equal contribution

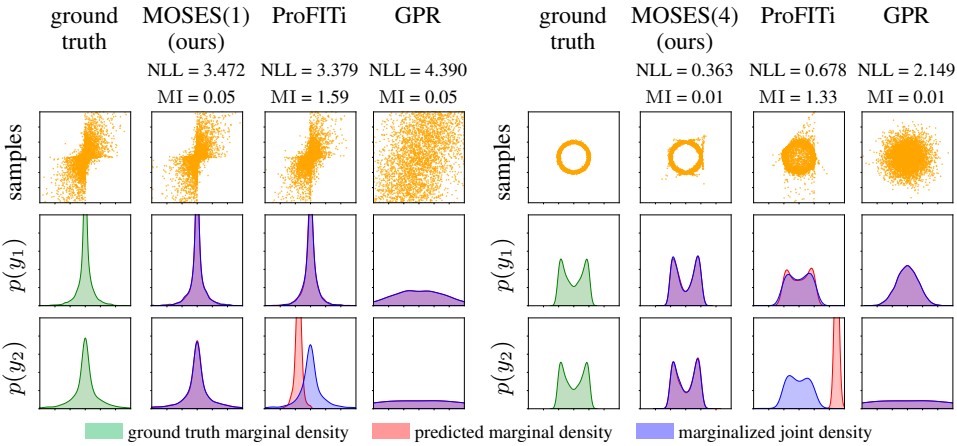

Figure 1: Demonstration of marginal consistency for MOSES (ours), ProFITi (Yalavarthi et al., 2025), and Gaussian Process Regression (GPR) (Bonilla et al., 2007) on two toy datasets: `blast` and `circle`. ProFITi is inconsistent w.r.t. the marginals of the second variable $y_2$, while $\text{MOSES}(D)$, where $D$ is the number of mixture components, is consistent with the marginals of both $y_1$ and $y_2$. GPR is marginalization consistent but predicts incorrect distributions. Marginalization inconsistency (MI, eq. 17) empirically estimates the KL divergence between the predicted marginal and the marginalized joint. See Appendix C.6 for more details.

**Example 1.1.** *In an ICU monitoring system, such a model might predict that blood pressure remains stable with 90% probability when queried individually, but only 60% when inferred from the joint distribution of vitals. These contradictions undermine reliability and could mislead clinical decisions.*

Due to inconsistency, ProFITi suffers significantly on marginal distributions despite having good joints. On the other hand, existing consistent approaches, such as Gaussian Process Regression (GPR) (Dürichen et al., 2015), avoid this problem but are limited to Gaussian distributions, creating a false contradiction between correctness and flexibility.

We address this problem with **MOSES (Mixtures of Separable Flows)**, a model based on mixtures of normalizing flows. Each component of MOSES uses a multivariate Gaussian, induced by a Gaussian process, as the base distribution, combined with univariate invertible transformations to modulate the marginals—similar in spirit to Gaussian Copula Processes (Wilson & Ghahramani, 2010). Since mixtures, Gaussian processes, and univariate transformations are each marginalization consistent, MOSES is consistent **by design**. At the same time, it retains the expressiveness of flow-based methods for the marginals.

Our contributions are as follows:

1. We formalize marginalization consistency as a fundamental requirement for probabilistic forecasting models and show that violating it renders a model incoherent.
2. We propose MOSES which achieves the expressiveness of modern flow approaches while maintaining strict marginalization consistency through a principled architectural design.
3. We show that consistency does not come at high performance cost: MOSES matches inconsistent models on joint prediction tasks while significantly outperforming them on marginal predictions, providing a principled solution to irregular time series forecasting.

## 2 PRELIMINARIES

Let $\text{Seq}(\mathcal{X})$ denote the space of finite sequences over $\mathcal{X}$. We represent an irregular time series $X$ as a sequence of $N$ triplets (Horn et al., 2020; Yalavarthi et al., 2025):

$$X = \left((t_n^{\text{OBS}}, c_n^{\text{OBS}}, v_n^{\text{OBS}})\right)_{n=1:N} \in \text{Seq}(\mathcal{X}), \quad \mathcal{X} = \mathbb{R} \times \{1, \ldots, C\} \times \mathbb{R}, \tag{1}$$

where $t_n^{\text{OBS}}$ is the observation time, $c_n^{\text{OBS}}$ the channel, and $v_n^{\text{OBS}}$ the observed value.

A **time series query** $Q$ is a sequence of $K$ pairs:

$$Q = \left((t_k^{\text{QRY}}, c_k^{\text{QRY}})\right)_{k=1:K} \in \text{Seq}(\mathcal{Q}), \quad \mathcal{Q} = \mathbb{R} \times \{1, \ldots, C\}, \tag{2}$$

where $t_k^{\text{QRY}}$ and $c_k^{\text{QRY}}$ specify the future time and queried channel.

A **forecasting answer** is a sequence $y = (y_1, \ldots, y_K)$ with $y_k \in \mathbb{R}$ predicting the value in channel $c_k^{\text{QRY}}$ at $t_k^{\text{QRY}}$. All query points occur after the observations: $\min_k t_k^{\text{QRY}} > \max_n t_n^{\text{OBS}}$.

**Requirements.** A marginalization consistent probabilistic irregularly sampled time series forecasting model must satisfy the following requirements:

**R1 Joint Multivariate Prediction.** The goal of probabilistic irregular time series forecasting is to model the joint distribution[1] $\hat{p}(y \mid Q, X)$ over responses $y$, given query points $Q$ and observed series $X$, allowing both context length $N = |X|$ and query length $K = |Q|$ to vary dynamically:

$$\begin{aligned}
\hat{p}\colon \text{Seq}(\mathbb{R} \times \mathcal{Q}) \times \text{Seq}(\mathcal{X}) &\longrightarrow \mathbb{R}_{\geq 0}, \\
(y, Q, X) &\longmapsto \hat{p}(y_1, \ldots, y_K \mid Q_1, \ldots, Q_K, X_1, \ldots, X_N)
\end{aligned} \tag{3}$$

For a given $(Q, X)$, the mapping $(y_1, \ldots, y_K) \mapsto \hat{p}(y_1, \ldots, y_K \mid Q, X)$ defines a probability density on $\mathbb{R}^{|Q|}$.

Unlike standard multivariate time series forecasting, where future time points are fixed and typically ignored, irregular time series require conditioning on future time points and channels, as these can vary across instances.

**R2 Permutation Invariance.** As the time stamp and channel-ID are included in each sample, the order of the samples does not matter, and hence any model prediction should be independent of the order of both the query or context:

$$\hat{p}(y \mid Q, X) = \hat{p}(y^{\pi} \mid Q^{\pi}, X^{\tau}) \quad \forall \pi \in \text{Sym}(|Q|), \tau \in \text{Sym}(|X|) \tag{4}$$

**R3 Marginalization Consistency/Projection Invariance.** Predicting the joint density for the sub-query $Q_{-k}$ given by removing the $k$-th item from $Q$ should yield the same result as marginalizing the $k$-th variable from the complete query $Q$.

$$\hat{p}(y_{-k} \mid Q_{-k}, X) = \int_{\mathbb{R}} \hat{p}(y \mid Q, X) \, dy_k \tag{5}$$

This can be extended to any subset $K_S \subseteq \{1, \ldots K\}$ via induction: Assuming **R3** holds whenever we marginalize out a set of variables $K_S$ with $|K_S| \leq m$, then, w.l.o.g., let $K_S' = \{1, \ldots, m+1\} = \{1, \ldots, m\} \cup \{m+1\}$. Now, simply marginalize out the first $m$ variables, and apply the property a second time to marginalize out $m+1$ as well (see Appendix A.1 for details).

For a model satisfying **R1**-**R3**, we will only have to marginalize if we try to validate the marginalization consistency. For this validation we added requirement **R3**. Yalavarthi et al. (2025) discussed **R1** and **R2**, but did not consider **R3**. We argue that irregularly sampled time series are realizations of a stochastic process and **R3** is a fundamental property of any model that mimics it.

**Theorem 2.1.** *Any model that satisfies **R1**-**R3** realizes an $\mathbb{R}$-valued stochastic process over the index set $T = \mathbb{R} \times \{1, \ldots, C\}$.*
*Proof. This is a direct application of Kolmogorov's extension theorem (Øksendal, 2003)*

WHY DO WE NEED MARGINALIZATION CONSISTENCY?

1. *Without marginalization consistency, probabilistic models can be unreliable.* Two test cases with the same context $X$ but different queries $Q$ may share overlapping targets. An inconsistent model can assign different distributions to these targets, despite identical conditioning information, which is unrealistic. Such models are inapplicable in many real-world applications.

---

[1]By abuse of notation, $p(y \mid t, X)$ denotes $p_{Y_t \mid X}(y \mid X)$. For $Q = (t_1, \ldots, t_n)$, $p(y \mid Q, X)$ denotes $p_{(Y_{t_1}, \ldots, Y_{t_n}) \mid X}(y_{t_1}, \ldots, y_{t_n} \mid X)$. Here "$\mid Q$" is index selection, not probabilistic conditioning on $Q$.

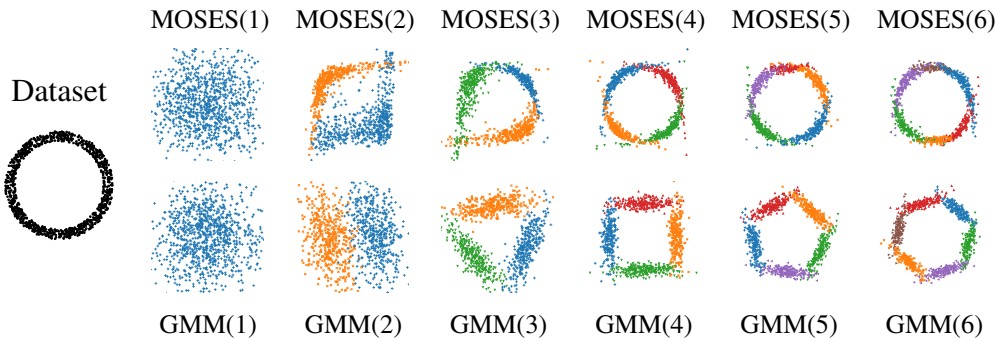

Figure 2: (Top) Importance of multiple mixture components: MOSES(1) cannot represent the correct distribution, but MOSES(4) can. (Bottom) Limitation of Gaussian Mixture Models: more components are needed as single ellipsoids cannot model curvature. Color represents mixture component; see Appendix C.6 or details.

2. *Marginalization consistency provides performance guarantees.* For a consistent model, if the joint prediction over $K$ points is accurate, predictions for any subset of size $< K$ are also guaranteed to be accurate. This follows from the **data processing inequality (DPI; Murphy, 2022)**, which ensures marginalizing cannot increase KL divergence:

$$D_{KL}\Big(p(y_1,\ldots,y_K \mid Q_1,\ldots,Q_K,X) \,\Big\|\, \hat{p}(y_1,\ldots,y_K \mid Q_1,\ldots,Q_K,X)\Big)$$

$$\geq D_{KL}\Big(p(y_1,\ldots,y_{K-1} \mid Q_1,\ldots,Q_{K-1},X) \,\Big\|\, \hat{p}(y_1,\ldots,y_{K-1} \mid Q_1,\ldots,Q_{K-1},X)\Big)$$

Intuitively, marginalization "averages out" errors, so accuracy cannot worsen on subsets. Consistent models thus remain reliable for marginal predictions, as confirmed in Tables 1 and 2, where inconsistent model, ProFITi, performs poorly on marginals despite good joint predictions.

## 3 CONSTRUCTING MARGINALIZATION CONSISTENT CONDITIONAL DISTRIBUTIONS

Our goal is to build a model for the conditional joint distribution $p(y_1,\ldots,y_K \mid Q_1,\ldots,Q_K,X)$, as in (3). Since the model should satisfy **R3**, it follows that the marginal distribution of $y_k$ must only depend on $Q_k$ and $X$.

**Separably Parametrized Gaussians.** A simple way to model a permutation-invariant conditional distribution for a variable number of targets is a multivariate Normal distribution $\mathcal{N}(y \mid \mu(X,Q), \Sigma(X,Q))$, where the mean and covariance are **separably parametrized**:

$$\mu_k = \tilde{\mu}(Q_k,X), \quad \Sigma_{k,\ell} = \widetilde{\Sigma}(Q_k,Q_\ell,X), \tag{6}$$

with functions $\tilde{\mu}$ and $\widetilde{\Sigma}$ as in Gaussian Processes. Such a setup is **marginalization consistent** by design, since marginalizing a Gaussian simply involves selecting the relevant rows and columns of $\Sigma$ and elements of $\mu$. However, Gaussian Processes are limited because they can only model joint Gaussian distributions. For more flexible distributions, **normalizing flows** are a common alternative (Rezende & Mohamed, 2015).

**Separable Normalizing Flows.** Normalizing flows model a distribution by applying an invertible transformation $f\colon \mathbb{R}^K \to \mathbb{R}^K$ to a source distribution $p_Z$ on $\mathbb{R}^K$. The density of the resulting target distribution is given by the change-of-variable formula:

$$p_Y(y) = p_Z(f^{-1}(y;\theta)) \cdot \left|\det\left(\frac{\partial f^{-1}(y;\theta)}{\partial y}\right)\right|. \tag{7}$$

Most existing flows use simple source distributions, typically a standard multivariate normal, and model interactions between variables through the transformation (Rezende & Mohamed, 2015;

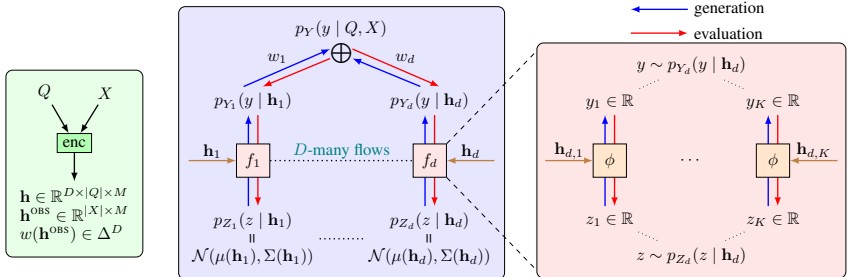

Figure 3: Illustration of MOSES. $D$ flows (fixed). $K$ variables (variable at inference time). The Encoder (enc) takes the observed series $X$ and query $Q$ as input, and outputs an embedding $\mathbf{h}$ (depends on both $X$ and $Q$) and $w$ (depends only on $X$). $\mathbf{h}_d$ is used to parametrize both $\mu_d, \Sigma_d$ of $p_{Z_d}$ and the flow transformation $f_d$. Each flow is separable and applies a spline transform $\phi$ to each component $z_k$ of $z \sim p_{Z_d}(z \mid \mathbf{h}_d)$, yielding $y_k$ of $y \sim p_d^{\text{FLOW}}(y \mid \mathbf{h}_d)$.

Papamakarios et al., 2021). Conditional flows for a variable number of targets follow the same idea, designing expressive transformations for vectors of arbitrary size (Liu et al., 2019; Biloš & Günnemann, 2021; Yalavarthi et al., 2025). However, these models generally **do not guarantee marginalization consistency**, and no simple condition on the transform alone can provide it.

We propose a different approach: use **simple, separable transformations** combined with a **richer source distribution**, namely a Gaussian Process with full covariance. In this setup, dependencies between variables are captured by the source distribution rather than the transformation, enabling marginalization consistency. This is similar in spirit to copula-based architectures, where dependencies between variables can be separated from their univariate marginal distributions.

**Lemma 3.1** (separable flow condition). *Assume each variable's transformation $\phi$ depends only on its own query and the shared context, not on other queries. A conditional flow model over $\mathbb{R}^K$ or $\mathrm{Seq}(\mathbb{R})$ is separable if it is expressed in the form*

$$f(z \mid Q, X) = \big(\phi(z_1 \mid Q_1, X), \ldots, \phi(z_K \mid Q_K, X)\big) \tag{8}$$

*for some univariate function $\phi\colon \mathbb{R} \times \mathcal{Q} \times \mathrm{Seq}(\mathcal{X}) \to \mathbb{R}$ that is invertible in the first argument. Any model that consists of such a separable flow transformation, combined with a marginalization consistent model for the source distribution, is itself marginalization consistent. (Proof: Appendix A.2)*

**Conditional Mixtures of Flows.** Using Gaussians as base distributions, we can model only linear dependencies between variables. To increase expressiveness, we combine multiple separable flows into a mixture. Even a few components can significantly improve the model, reaching performance comparable to a simple GMM without flow transformations (Figure 2, Appendix C.6).

**Lemma 3.2.** *Given probabilistic models $(\hat{p}_d)_{d=1:D}$ that satisfy **R1-R3**, then a mixture model*

$$\hat{p}(y \mid Q, X) = \sum_{d=1}^{D} w_d(X)\, \hat{p}_d(y \mid Q, X) \tag{9}$$

*with permutation invariant weight function $w\colon \mathrm{Seq}(\mathcal{X}) \to \Delta^D$, where $\Delta^D$ denotes probability simplex in $D$ variables $\Delta^D := \{w \in \mathbb{R}^D \mid w_d \geq 0, \sum_d w_d = 1\}$, also satisfies **R1-R3**. (Proof: Appendix A.3)*

## 4   MIXTURES OF SEPARABLE FLOWS (MOSES)

Based on the constructions from the previous section, we build a marginalization-consistent model for forecasting irregular time series using four components (Figure 3):

1. **Separable encoder:** A *shared encoding* of the observations, $\mathbf{h}^{\text{OBS}} := \mathrm{enc}^{\text{OBS}}(X; \theta^{\text{OBS}})$, and $D$ vectorized, *query-specific* encodings $\mathbf{h}_{d,k} := \mathrm{enc}_d^{\text{QRY}}(Q_k, X; \theta_d^{\text{QRY}})$.

2. **Separable Gaussians:** $D$ multivariate Gaussians, $p_{Z_d}(z \mid \mu_d, \Sigma_d)$, each parametrized separately by the corresponding query encoding $\mathbf{h}_d$.

3. **Separable transformations:** $D$ separable normalizing flows $f_d$ are parametrized using the encoded queries $\mathbf{h}_d$. Each $f_d$ is applied on top of the Gaussian base distribution $p_{Z_d}$, providing a separable normalizing flow $\hat{p}_d^{\text{FLOW}}$.

4. **Mixture of flows:** The $D$ flows are combined using mixing weights $w := w(\mathbf{h}^{\text{OBS}})$, which only depend on the shared observation encoding $\mathbf{h}^{\text{OBS}}$, not on the queries.

**1. Separable Encoder.** To encode both the observations $X = ((t_n^{\text{OBS}}, c_n^{\text{OBS}}, v_n^{\text{OBS}}))_{n=1:N}$ and queries $Q = ((t_k^{\text{QRY}}, c_k^{\text{QRY}}))_{k=1:K}$, we apply a positional embedding with learnable parameters $(a_f, b_f)_{f=1:F}$ to the time component (Kazemi et al., 2019).

$$\text{pos\_embed}(t)_f := \begin{cases} a_f t + b_f & \text{if } f = 1 \\ \sin(a_f t + b_f) & \text{else} \end{cases} \tag{10}$$

We use one-hot encodings for the channel component. The value component is passed through.

$$\mathbf{x} := \left[ \text{pos\_embed}(t_n^{\text{OBS}}), \text{one-hot}(c_n^{\text{OBS}}), v_n^{\text{OBS}} \right]_{n=1:N} \tag{11a}$$

$$\mathbf{q} := \left[ \text{pos\_embed}(t_k^{\text{QRY}}), \text{one-hot}(c_k^{\text{QRY}}) \right]_{k=1:K} \tag{11b}$$

The observations $\mathbf{x} \in \mathbb{R}^{N \times (F+C+1)}$ are further encoded via self-attention, and the queries $\mathbf{q} \in \mathbb{R}^{K \times (F+C)}$ via cross-attention w.r.t. the encoded observations:

$$\mathbf{h}^{\text{OBS}} := \text{MHA}(\mathbf{x}, \mathbf{x}, \mathbf{x}; \theta^{\text{OBS}}) \qquad (\in \mathbb{R}^{N \times M}) \tag{12a}$$

$$\widetilde{\mathbf{h}} := \text{MHA}(\mathbf{q}, \mathbf{h}^{\text{OBS}}, \mathbf{h}^{\text{OBS}}; \theta^{\text{QRY}}) \quad (\in \mathbb{R}^{K \times D \cdot M}) \tag{12b}$$

$$\mathbf{h} := \text{reshape}(\widetilde{\mathbf{h}}) \qquad (\in \mathbb{R}^{D \times K \times M}) \tag{12c}$$

where MHA denotes multi-head attention. For the encoding of the queries, we use an encoding dimension $D \cdot M$ and reshape each $\mathbf{h}_k$ into $D$ encodings $\mathbf{h}_{d,k}$ of dimension $M$.

**2. Separable Gaussians $p_{Z_d}(z \mid \mu_d, \Sigma_d)$.** We model the means and covariances of the base multivariate Gaussian using simple linear and quadratic functions of the encoded queries $\mathbf{h}_d$:

$$\mu(\mathbf{h}_d) = \mathbf{h}_d \theta^{\text{MEAN}} \qquad \Longrightarrow \quad \mu(\mathbf{h}_d)_k = \mathbf{h}_{d,k} \theta^{\text{MEAN}}, \tag{13a}$$

$$\Sigma(\mathbf{h}_d) = \mathbb{I}_K + \frac{(\mathbf{h}_d \theta^{\text{COV}})(\mathbf{h}_d \theta^{\text{COV}})^T}{\sqrt{M'}} \quad \Longrightarrow \quad \Sigma(\mathbf{h}_d)_{k,l} = \delta_{kl} + \frac{(\mathbf{h}_{d,k} \theta^{\text{COV}})(\mathbf{h}_{d,l} \theta^{\text{COV}})^T}{\sqrt{M'}}, \tag{13b}$$

where $\theta^{\text{MEAN}} \in \mathbb{R}^{M \times 1}$ and $\theta^{\text{COV}} \in \mathbb{R}^{M \times M'}$ are trainable weights shared across all $D$ mixture components, $\mathbb{I}_K$ is the $K \times K$ identity matrix, and $\delta_{kl}$ is the Kronecker delta.

The scaling by $\sqrt{M'}$ in (13b) ensures stable learning, following (Vaswani et al., 2017). Since $\Sigma(\mathbf{h}_d)$ is the sum of a positive semidefinite and a positive definite matrix, it remains positive definite. The encoding $\mathbf{h}_d$ incorporates both the context $X$ and the queries $Q$, consistent with the separable Gaussian setup in (6). Note that although the base Gaussian is defined via Gaussian processes (GPs), we do not perform GP-style inference.

**3. Separable transformations $f_d$.** To obtain separable invertible transformations, we apply a univariate bijective function to each variable independently. Spline-based functions are particularly popular due to their expressiveness and good generalization (Durkan et al., 2019; Dolatabadi et al., 2020). We use computationally efficient Linear Rational Spline (LRS) transformations (Dolatabadi et al., 2020). For a conditional LRS $\phi(z_k \mid \mathbf{h}_{d,k}; \theta^{\text{FLOW}})$, parameters such as bin widths and heights, derivatives at the knots, and $\lambda$ are computed from the conditioning input $\mathbf{h}_{d,k}$ and shared model parameters $\theta^{\text{FLOW}}$. The same $\theta^{\text{FLOW}}$ is used for all variables $z_{1:K}$, allowing the transformation to handle varying numbers of variables, and is also shared across all $D$ mixture components. For details, see Appendix A.5. In summary, the conditional flow model is separable across the query size $f = f_1 \times \cdots \times f_K$ with

$$f_d(z \mid Q, X) := f(z \mid \mathbf{h}_d) = \left( \phi(z_1, \mathbf{h}_{d,1}), \ldots, \phi(z_K, \mathbf{h}_{d,K}) \right) \tag{14}$$

**4. Mixture Model.** We model the mixture weights via cross-attention, using trainable parameters $\beta \in \mathbb{R}^{D \times M}$ as attention queries, and a softmax to ensure the weights sum to 1:

$$w := \text{softmax}(\text{MHA}(\beta, \mathbf{h}^{\text{OBS}}, \mathbf{h}^{\text{OBS}}; \theta^{\text{MIX}})) \tag{15}$$

**Theorem 4.1.** *Our model, MOSES, satisfies **R1**-**R3** and hence realizes a stochastic process via Kolmogorov's Extension Theorem (see Theorem 2.1). Proof: Appendix A.4.*

**Computational Complexities.** The $D$ separable flows are computationally efficient: since they are separable, their Jacobian matrix is diagonal, and computing its determinant requires $\mathcal{O}(K)$ operations. The main computational cost lies in evaluating $\Sigma_d^{-1}$ and $\det \Sigma_d^{-1}$ for the base distribution, which typically requires $\mathcal{O}(K^3)$ operations. However, for large $K$, our low-rank modification $\Sigma_d = \mathbb{I}_K + UU^T$ (see (13b)) reduces their computation to $\mathcal{O}(M'^2 K)$ using the Woodbury and Weinstein–Aronszajn identities. This approach scales well for large values of $K \gg M'$, as $M'$ is independent of $K$.

**Training.** Given a batch $\mathcal{B}$ of training instances $(Q, X, y)$, we minimize the normalized joint negative log-likelihood (njNLL) (Yalavarthi et al., 2025):

$$\mathcal{L}^{\text{njNLL}}(\theta) = \frac{1}{|\mathcal{B}|} \sum_{(Q,X,y) \in \mathcal{B}} -\frac{1}{|y|} \log \hat{p}(y \mid Q, X) \tag{16}$$

where $\theta$ includes all the model parameters. njNLL generalizes NLL to varying target sizes.

**Relationship to Copulas.** Each component of MOSES is conceptually similar to a Gaussian copula process Wilson & Ghahramani (2010), in that it separates the linear dependency structure from the marginal distributions. The base Gaussian covariance matrix can be decomposed into a correlation matrix and diagonal variances. The diagonal variances, together with the base mean and the univariate spline transformations, determine the contribution of each component to the marginal distribution of each variable. On the other hand, since MOSES is a mixture of $D$ such components, each with its own transformations, the overall model does not have a simple separation between dependency structure and marginal distributions.

Many copula-based architectures train marginals and dependencies separately to avoid identifiability issues. However, MOSES is trained end-to-end, and because of the restricted domain of the spline transformations, identifiability issues are largely mitigated. A shift in the mean of the base distribution cannot be compensated by the spline transformation.

To the best of our knowledge, copula-based architectures have not been applied to probabilistic forecasting of irregular time series. Models such as TACTiS and TACTiS-2 (Drouin et al., 2022; Ashok et al., 2024) are designed for regularly sampled, fully observed data. While their principles could extend to irregular time series, their implementations are not readily adaptable, and they lack marginalization consistency due to their non-separable encoder and copula structure.

## 5 RELATED WORK

There have been multiple works that deal with point forecasting of irregular time series (Ansari et al., 2023; Che et al., 2018; Scholz et al., 2023; Chen et al., 2024; Yalavarthi et al., 2024; Zhang et al., 2024; Klötergens et al., 2024; 2025a;b) In this work, we deal with probabilistic forecasting of irregular time series. Models such as NeuralFlows (Biloš et al., 2021), GRU-ODE (De Brouwer et al., 2019), and CRU (Schirmer et al., 2022) predict only the marginal distribution for a single time stamp. Additionally, interpolation models like HetVAE (Shukla & Marlin, 2022) and Triplet-former (Yalavarthi et al., 2023) can also be applied for probabilistic forecasting. However, they also produce only marginal distributions. All the above models assume the underlying distribution is Gaussian which is not the case for many real-world datasets. On the other hand, Gaussian Process Regression (GPR Dürichen et al., 2015), and ProFITi (Yalavarthi et al., 2025) can predict proper joint distributions. ProFITi is not marginalization consistent because of its non-separable encoder and probabilistic component.

There have been works on models for tractable and consistent marginals for fixed size variables (e.g. tabular data). Probabilistic Circuits (Choi et al., 2020) create a sum-prod network on the marginal distributions. Later, Sidheekh et al. (2023) added univariate normalizing flows to the leaf nodes of the circuit for better expressiveness. However, it is not trivial to extend such circuits to deal with sequential data of variable size. Gaussian Mixture Models (GMMs) (Duda & Hart, 1974) are often used only for unconditional density estimation, but can be extended to conditional density estimation. They can provide tractable and consistent marginal distributions. However, GMMs are not expressive enough and often require a very large number of components to approximate even

Table 1: Comparing njNLL. Lower is better, best results in bold, second best in italics.

|  | Model | USHCN | PhysioNet'12 | MIMIC-III | MIMIC-IV |
|---|---|---|---|---|---|
| inconsistent | ProFITi | *-3.226 ± 0.225* | **-0.647 ± 0.078** | **-0.377 ± 0.032** | **-1.777 ± 0.066** |
| consistent univariate | GRU-ODE | 0.766 ± 0.159 | 0.501 ± 0.001 | 0.961 ± 0.064 | 0.823 ± 0.318 |
|  | NeuralFlows | 0.775 ± 0.152 | 0.496 ± 0.001 | 0.998 ± 0.177 | 0.689 ± 0.087 |
|  | CRU | 0.761 ± 0.191 | 1.057 ± 0.007 | 1.234 ± 0.076 | OOM |
|  | Tripletformer+ | 4.632 ± 8.179 | 0.519 ± 0.112 | 1.051 ± 0.141 | 0.686 ± 0.115 |
| consistent multivariate | GPR | 2.011 ± 1.376 | 1.367 ± 0.074 | 3.146 ± 0.359 | 2.789 ± 0.057 |
|  | GMM | 1.050 ± 0.031 | 1.063 ± 0.002 | 1.160 ± 0.020 | 1.076 ± 0.003 |
|  | MOSES (ours) | **-3.357 ± 0.176** | *-0.491 ± 0.041* | *-0.305 ± 0.027* | *-1.668 ± 0.097* |

simple distributions, see Figure 2. Note that normalizing flow models such as Dinh et al. (2017); Papamakarios et al. (2017; 2021) neither provide tractable marginals nor are applicable to varying number of variables.

Prior work on mixtures of normalizing flows has focused on fixed-length sequences. Pires & Figueiredo (2020) and Ciobanu (2021) used affine coupling and masked autoregressive flows for density estimation, while Postels et al. (2021) applied them to reconstruction tasks. These models face challenges with variable-length sequences and intractable marginals. Sendera et al. (2021) introduced non-Gaussian Gaussian Processes for few-shot learning but only support single-variable prediction. In contrast, MOSES handles multiple variables and missing values.

## 6 EXPERIMENTS

### 6.1 TOY EXPERIMENT

We illustrate marginalization consistency using two synthetic bivariate distributions (Blast and Circle; see Figure 1, Appendix B). The task is to estimate the unconditional joint distribution. MOSES accurately models both joint and marginal distributions while preserving consistency. In contrast, ProFITi captures the joint well — especially for Blast — but fails on marginals due to its triangular attention mechanism, which enforces a fixed dependency order. Gaussian Process Regression maintains consistency but lacks predictive accuracy. To predict the marginalization inconsistency we use **2-Wasserstein distance (WD)**. For each variable $y_k$, we compare:

1. $\hat{p}(y_k \mid Q_k, X)$: the predicted marginal,
2. $\hat{p}^{\text{MAR}}(y_k \mid Q_k, X)$: the marginal obtained by integrating the joint $\hat{p}(y \mid Q, X)$.

Since sampling directly from $\hat{p}^{\text{MAR}}$ is difficult, we sample from the joint and extract the $k$-th component. The marginalization inconsistency is defined as the average WD across all $K$ variables:

$$\text{MI} = \frac{1}{K} \sum_{k=1}^{K} \text{WD}\Big(\hat{p}(y_k \mid Q_k, X), \hat{p}^{\text{MAR}}(y_k \mid Q_k, X)\Big). \tag{17}$$

We use 1000 samples to compute MI. Currently, we compute only univariate marginals; multivariate marginals are possible in principle but computationally prohibitive.

### 6.2 MAIN EXPERIMENT

We evaluate our model on four real-world datasets: one climate dataset (USHCN) and three medical datasets (PhysioNet'12, MIMIC-III, and MIMIC-IV). Following prior work (Yalavarthi et al., 2025; Biloš et al., 2021), we observe the first 36h and predict the next 3 time steps for medical datasets, and observe 3 years and predict 3 time steps for USHCN. Both the number of observations ($N$) and queries ($K$) vary across samples (see Table 5). We split each dataset into training, validation, and test sets using a 70:10:20 ratio. We train MOSES using the Adam optimizer with a learning rate of 0.001 and a batch size of 64. Hyperparameter search is over mixture components $D \in \{1, 3, 5, 7, 10\}$,

Table 2: Trained for njNLL and evaluate for mNLL, lower is better.

|  | Model | USHCN | PhysioNet'12 | MIMIC-III | MIMIC-IV |
|---|---|---|---|---|---|
| inconsistent | ProFITi | *-3.324 ± 0.206* | *-0.016 ± 0.085* | *0.408 ± 0.030* | *0.500 ± 0.322* |
| consistent univariate | GRU-ODE | 0.776 ± 0.172 | 0.504 ± 0.061 | 0.839 ± 0.030 | 0.876 ± 0.589 |
|  | Neural-Flows | 0.775 ± 0.180 | 0.492 ± 0.029 | 0.866 ± 0.097 | 0.796 ± 0.053 |
|  | CRU | 0.762 ± 0.180 | 0.931 ± 0.019 | 1.209 ± 0.044 | OOM |
|  | Tripletformer+ | 0.411 ± 7.506 | 0.524 ± 0.110 | 0.894 ± 0.083 | 0.751 ± 0.063 |
| consistent multivariate | GPR | 1.235 ± 0.096 | 1.161 ± 0.065 | 1.341 ± 0.009 | 1.161 ± 0.010 |
|  | GMM | 1.042 ± 0.021 | 1.069 ± 0.002 | 1.124 ± 0.007 | 1.075 ± 0.007 |
|  | MOSES (ours) | **-3.355 ± 0.156** | **-0.271 ± 0.028** | **0.163 ± 0.026** | **-0.634 ± 0.017** |

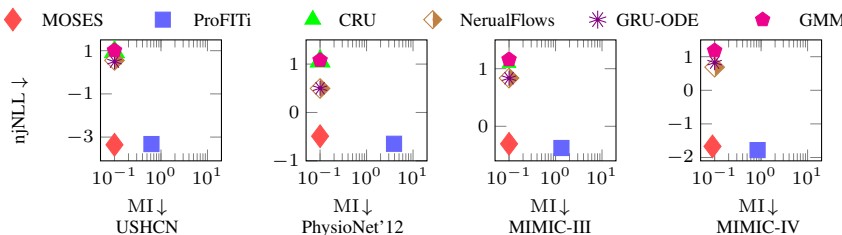

Figure 4: njNLL vs. MI. MOSES is marginalization consistent within sampling error.

attention heads $\in \{1, 2, 4\}$, and latent sizes $M, F \in \{16, 32, 64, 128\}$. All models are implemented in PyTorch and trained on NVIDIA RTX 3090, A40 and GTX 1080 Ti GPUs.

**Baselines.** We use NeuralFlows (Biloš et al., 2021), GRU-ODE (De Brouwer et al., 2019), CRU (Schirmer et al., 2022), GPR (Dürichen et al., 2015), and ProFITi (Yalavarthi et al., 2025). Our encoder is similar to Tripletformer (Yalavarthi et al., 2023) that predict marginal distributions for interpolation. We used it for forecasting and called the model Tripletformer+. NeuralFlows, GRU-ODE, CRU, and Tripletformer+ predict only marginals and are marginalization consistent, as their joint distribution is the product of marginals. GPR is also marginalization consistent. We also compare with Gaussian Mixture Model (GMM) which is MOSES without flows attached to highlight the advantage of flows in MOSES.

**Results.** To highlight the importance of Marginalization Consistency in probabilistic forecasting models we train the model for njNLL (Equation (16)) and evaluate for two metrics: 1. Normalized Joint Negative Log-Likelihood (njNLL; Table 1) and 2. Marginal Negative Log-Likelihood (mNLL; Table 2). While njNLL measures joint density, mNLL assesses univariate marginal density (Biloš et al., 2021; Schirmer et al., 2022). A good model must perform well on both the metrics.

MOSES outperforms all marginalization-consistent models across both metrics. Although it performs similarly or slightly worse than ProFITi on njNLL, it significantly surpasses ProFITi on mNLL. For USHCN, both models perform comparably within standard deviation. From Figure 4, our model achieves similar likelihoods to ProFITi, with MI values close to zero, while MI values of ProFITi are up to an order of magnitude larger. This difference arises because ProFITi prioritizes joint distributions but neglects marginalization consistency, leading to performance degradation when queried on single instances (Figure 1). In contrast, our model maintains consistency. Similar results are observed when comparing Energy Score (for multivariate distributions) and CRPS (for univariate distributions) as seen in Table 3.

We noticed that ProFITi's gains primarily stem from its encoder. When using the same encoder (ProFITi-TF), MOSES achieves superior accuracy in MIMIC-III and MIMIC-IV (Table 4). Furthermore, when we replaced our encoder with the encoder of ProFITi which is GraFITi, MOSES-GraFITi has significantly better results than ProFITi in both MIMIC-III and MIMIC-IV. However, since GraFITi is an inconsistent model, MOSES-GraFITi has worse mNLL for all the datasets.

Table 3: Comparing models w.r.t. Energy Score and CRPS. Lower is better, best results in bold.

| Dataset | Energy Score | | CRPS | |
|---|---|---|---|---|
| | ProFITi | MOSES (ours) | ProFITi | MOSES (ours) |
| USHCN | **0.452 ± 0.044** | 0.552 ± 0.044 | **0.182 ± 0.007** | 0.220 ± 0.019 |
| PhysioNet'12 | **0.879 ± 0.303** | 1.599 ± 0.013 | 0.271 ± 0.003 | **0.260 ± 0.002** |
| MIMIC-III | 1.606 ± 0.168 | **1.353 ± 0.033** | 0.319 ± 0.003 | **0.296 ± 0.005** |
| MIMIC-IV | **0.808 ± 0.003** | 0.906 ± 0.029 | 0.279 ± 0.012 | **0.245 ± 0.010** |

Table 4: Comparing njNLL and mNLL across datasets to verify the contribution of probabilistic component of ProFITi. "ProFITi-TF" denotes ProFITi-Transformer using same encoder as MOSES. "MOSES-GraFITi" is MOSES with GraFITi as encoder.

| | Dataset | | | |
|---|---|---|---|---|
| | USHCN | PhysioNet'12 | MIMIC-III | MIMIC-IV |
| | **njNLL** | | | |
| ProFITi | -3.226±0.225 | -0.647±0.078 | -0.377±0.032 | -1.777±0.066 |
| ProFITi-TF | **-3.415±0.271** | **-0.657±0.034** | 0.516±0.111 | -1.405±0.220 |
| MOSES-GraFITi | -3.282±0.260 | -0.615±0.030 | **-0.396±0.025** | **-2.197±0.027** |
| MOSES | -3.357±0.176 | -0.491±0.041 | -0.305±0.027 | -1.668±0.097 |
| | **mNLL** | | | |
| ProFITi | -3.324±0.206 | -0.016±0.085 | 0.408±0.030 | 0.500±0.322 |
| ProFITi-TF | **-3.440±0.243** | 0.017±0.042 | 1.279±0.057 | 0.345±0.325 |
| MOSES-GraFITi | -3.252±0.231 | -0.227±0.119 | 0.520±0.125 | -0.299±0.114 |
| MOSES | -3.355±0.156 | **-0.271±0.028** | **0.163±0.026** | **-0.634±0.017** |

**Discussion.** While MOSES provides strong theoretical guarantees through marginalization consistency, this comes with an inherent trade-off in expressivity. The separability constraint necessary for tractable marginalization limits the model's ability to capture complex dependencies directly through the flow transformations. In MOSES, dependencies are primarily captured through the low-rank covariance structure of latent Gaussian distribution and mixtures, while the expressive power of normalizing flows is focused on modeling flexible marginal distributions.

This architectural choice explains why MOSES achieves superior marginal performance and perfect consistency, while ProFITi—with its fully non-separable flows—can sometimes achieve better joint distribution modeling at the cost of consistency. In applications like healthcare forecasting, where reliability and interpretability are crucial, the consistency guarantees of MOSES provide great value despite this trade-off.

In future work, we aim to explore more flexible architectures that maintain marginalization consistency, such as through copula models or probabilistic circuits. A key challenge will be adapting these methods to handle the irregular time series setting.

## CONCLUSIONS

We introduced MOSES, a marginalization-consistent mixture of separable flows for probabilistic forecasting of irregular time series with missing values. By carefully parametrizing its components, we ensured both decomposability and marginalization consistency. Experiments on four real-world irregularly sampled datasets show that MOSES achieves substantial gains in marginal predictions while preserving competitive accuracy in joint predictions compared to more flexible but inconsistent models. This work provides an initial step toward addressing marginalization inconsistency in probabilistic forecasting. Future research will likely focus on enhancing performance while preserving consistency. A key limitation of MOSES is that mixture weights are independent of the query $Q$. Although adapting weights to query time points could improve joint predictions, requirements **R1**–**R3** enforce query-independence as a necessary condition for marginalization consistency.

## USAGE OF LARGE LANGUAGE MODELS

We would like to note that all scientific contributions in this manuscript such as conceptual ideas, problem identifications, model development, theorems, proofs, experimental results, discussion and so on are entirely the work of the authors. The usage of Large Language Models is limited to minor linguistic refinement such as correcting grammar and typos.

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

# A  THEORY

## A.1  PROOF OF EXTENSION OF EQUATION (5) TO ARBITRARY FINITE SETS

We show that Equation (5) is equivalent to the more general statement that for any finite subset $K_S \subseteq \{1, \ldots, K\}$ holds:

$$\hat{p}(y_{-K_S} \mid Q_{-K_S}, X) = \int_{\mathbb{R}^{|K_S|}} \hat{p}(y \mid Q, X) \, \mathrm{d}y_{K_S} \tag{18}$$

*Proof.* Obviously, (18) is equivalent to (5) when $|K_S| = 1$ via $K_S = \{k\}$. The other direction can be shown by induction on the size of $K_S$. Assume that (18) holds for any subset of size at most $m$, and let $K_S$ be a subset of size $m + 1$. W.l.o.g., we may assume that $K_S = \{1, \ldots, m+1\} = \{1, \ldots, m\} \cup \{m+1\}$. Let $\tilde{y} = y_{-\{1,\ldots,m\}} = (y_{m+1}, \ldots, y_K)$ and $\tilde{Q} = Q_{-\{1,\ldots,m\}} = (Q_{m+1}, \ldots, Q_K)$, then $y_{-K_S} = \tilde{y}_{-1}$ and $Q_{-K_S} = \tilde{Q}_{-1}$. It follows that

$$
\begin{aligned}
\hat{p}(y_{-K_S} \mid Q_{-K_S}, X) &= \hat{p}(\tilde{y}_{-1} \mid \tilde{Q}_{-1}, X) \\
&= \int \hat{p}(y_{m+1}, \ldots, y_K \mid Q_{m+1}, \ldots, Q_K, X) \, \mathrm{d}y_{m+1} \\
&= \int \left( \int \hat{p}(y_{1:K} \mid Q_{1:K}, X) \, \mathrm{d}y_{1,\ldots,m} \right) \mathrm{d}y_{m+1} \\
&= \int \hat{p}(y_{1:K} \mid Q_{1:K}, X) \, \mathrm{d}y_{1:m+1}
\end{aligned}
$$

$\square$

## A.2  PROOF OF LEMMA 3.1

*Proof.* Since $X$ is a common conditional to all the marginals, we can ignore it. So, assume that $f$ is a separable transformation:

$$f(z \mid Q) = (\phi(z_1 \mid Q_1), \ldots, \phi(z_K \mid Q_K)) \tag{19}$$

and that $\hat{p}_Z(z \mid Q)$ is marginalization consistent model. Then, the predictive distribution is

$$\hat{p}(y \mid Q) = \hat{p}_Z(f^{-1}(y \mid Q) \mid Q) \cdot \left| \det \frac{\mathrm{d}f^{-1}(y \mid Q)}{\mathrm{d}y} \right| \tag{20}$$

Since $f$ is separable, it follows that the Jacobian is diagonal:

$$
\begin{aligned}
\frac{\mathrm{d}f^{-1}(y \mid Q)}{\mathrm{d}y} &= \frac{\mathrm{d}\left( \phi^{-1}(y_1 \mid Q_1), \ldots \phi^{-1}(y_K \mid Q_K) \right)}{\mathrm{d}(y_1, \ldots, y_K)} \\
&= \mathrm{diag}\left( \frac{\mathrm{d}\phi^{-1}(y_1 \mid Q_1)}{\mathrm{d}y_1}, \ldots, \frac{\mathrm{d}\phi^{-1}(y_K \mid Q_K)}{\mathrm{d}y_K} \right)
\end{aligned}
\tag{21}
$$

Hence, the determinant of the Jacobian is the product of the diagonal elements:

$$\left| \det \frac{\mathrm{d}f^{-1}(y \mid Q)}{\mathrm{d}y} \right| = \prod_{k=1:K} \left| \det \frac{\mathrm{d}\phi^{-1}(y_k \mid Q_k)}{\mathrm{d}y_k} \right| = \prod_{k=1:K} \left| \frac{\mathrm{d}\phi^{-1}(y_k \mid Q_k)}{\mathrm{d}y_k} \right| \tag{22}$$

Using this fact, we can integrate the joint density over $y_k$ to get the marginal density:

$$\int \hat{p}(y \mid Q)\,\mathrm{d}y_k$$

$$= \int \hat{p}_Z\big(f^{-1}(y \mid Q) \mid Q\big) \cdot \left|\det \frac{\mathrm{d}f^{-1}(y \mid Q)}{\mathrm{d}y}\right| \mathrm{d}y_k \qquad\qquad \rhd (7)$$

$$= \int \hat{p}_Z\big(f^{-1}(y \mid Q) \mid Q\big) \cdot \prod_{k=1:K}\left|\frac{\mathrm{d}\phi^{-1}(y_k \mid Q_k)}{\mathrm{d}y_k}\right| \mathrm{d}y_k \qquad\qquad \rhd (22)$$

$$= \left(\prod_{l\neq k}\left|\frac{\mathrm{d}\phi^{-1}(y_l \mid Q_l)}{\mathrm{d}y_l}\right|\right) \cdot \int \hat{p}_Z\big(f^{-1}(y \mid Q) \mid Q\big) \cdot \left|\frac{\mathrm{d}\phi^{-1}(y_k \mid Q_k)}{\mathrm{d}y_k}\right| \mathrm{d}y_k$$

$$= \left(\prod_{l\neq k}\left|\frac{\mathrm{d}\phi^{-1}(y_l \mid Q_l)}{\mathrm{d}y_l}\right|\right) \cdot \int \hat{p}_Z(z \mid Q)\,\mathrm{d}z_k \qquad\qquad \rhd \text{transf.-thm}$$

$$= \left(\prod_{l\neq k}\left|\frac{\mathrm{d}\phi^{-1}(y_l \mid Q_l)}{\mathrm{d}y_l}\right|\right) \hat{p}_Z(z_{-k} \mid Q_{-k}) \qquad\qquad \rhd (5)$$

$$= \hat{p}_Z(z_{-k} \mid Q_{-k})\left|\det \frac{\mathrm{d}f^{-1}(y_{-k} \mid Q_{-k})}{\mathrm{d}y_{-k}}\right| \qquad\qquad \rhd (22)$$

$$= \hat{p}(y_{-k} \mid Q_{-k}) \qquad\qquad \rhd (7)$$

$$\square$$

## A.3 Proof of Lemma 3.2

*Proof.* Consider a mixture model of the form

$$\hat{p}(y \mid Q, X) := \sum_{d=1}^{D} w_d(X)\hat{p}_d(y \mid Q, X) \tag{23}$$

satisfying the conditions from Lemma 3.2, i.e. the component models $\hat{p}_d$ satisfy the requirements **R1**-**R3** and the weight function $w\colon \mathrm{Seq}(\mathcal{X}) \to \Delta^D$ is permutation invariant with respect to $X$.

1. $\hat{p}$ satisfies **R1**: By construction of the mixture model, it has the same domain and codomain as the component models.

2. $\hat{p}$ satisfies **R2**: Let $\pi \in \mathrm{Sym}(|Q|)$ and $\tau \in \mathrm{Sym}(|X|)$, then

$$\hat{p}(y^{\pi} \mid Q^{\pi}, X^{\tau}) = \sum_{d=1}^{D} w_d(X^{\tau})\hat{p}_d(y^{\pi} \mid Q^{\pi}, X^{\tau})$$

$$= \sum_{d=1}^{D} w_d(X)\hat{p}_d(y \mid Q, X) \qquad \rhd \text{permutation invariance of } w \text{ and } \hat{p}_d$$

$$= \hat{p}(y \mid Q, X)$$

3. $\hat{p}$ satisfies **R3**:

$$\int \hat{p}(y \mid Q, X)\,\mathrm{d}y_k = \int \sum_{d=1}^{D} w_d(X)\hat{p}_d(y \mid Q, X)\,\mathrm{d}y_k$$

$$= \sum_{d=1}^{D} w_d(X)\int \hat{p}_d(y \mid Q, X)\,\mathrm{d}y_k$$

$$= \sum_{d=1}^{D} w_d(X)\hat{p}_d(y_{-k} \mid Q_{-k}, X) \quad \rhd \hat{p}_d \text{ is marginalization consistent}$$

$$= \hat{p}(y_{-k} \mid Q_{-k}, X)$$

$\square$

## A.4 PROOF OF THEOREM 4.1

*Proof.* Due to Lemma 3.2, it is sufficient to show that all the component models satisfy the requirements **R1**-**R3**, and that the mixture weights are permutation invariant with respect to $X$.

The latter immediately follows from (12a) and (15): since $\mathbf{h}^{\text{OBS}}$ is computed by self-attention over $\mathbf{x}$, the encoding of $X$, it is permutation equivariant with respect to the sequence elements, i.e.

$$\mathbf{h}^{\text{OBS}}(\mathbf{x}^\tau) = (\mathbf{h}^{\text{OBS}}(\mathbf{x}))^\tau \quad \text{for any} \quad \tau \in \text{Sym}(|X|) \tag{24}$$

Then, $w$ is computed by cross-attention over $\mathbf{h}^{\text{OBS}}$, and hence is permutation invariant with respect to $X$, as the sequence length is summed over in Equation (15).

**R1**, joint multivariate prediction, is by construction.

**R3**, marginalization consistency, is satisfied by the component models due to Lemma 3.1, since we use separable Gaussians as source distributions and separable flows as transformations.

**R2**, permutation invariance of the component model $p_d$ can be seen as follows:

As already mentioned, $\mathbf{h}^{\text{OBS}}$ is permutation equivariant with respect to $X$. By Equation (12b), it follows that $\mathbf{h}_d$ is permutation invariant with respect to $X$, since the observation length is summed over, and permutation equivariant with respect to $Q$:

$$\mathbf{h}_d(Q^\pi, X^\tau) = \mathbf{h}_d(Q, X)^\pi \quad \text{for any} \quad \pi \in \text{Sym}(|Q|), \tau \in \text{Sym}(|X|) \tag{25}$$

Now, consider the $d$-th component model $\hat{p}_{Y_d}(y \mid Q, X)$. Let $z = f_d^{-1}(y \mid Q, X)$. By Equation (14), and the equivariance/invariance of $\mathbf{h}_d$ w.r.t. $Q$ and $X$, our flow model satisfies

$$f_d^{-1}(y^\pi \mid Q^\pi, X^\tau) = f^{-1}(y^\pi \mid \mathbf{h}_d^\pi) = (f^{-1}(y \mid \mathbf{h}_d))^\pi = z^\pi \tag{26}$$

I.e. $z$ is permutation equivariant with respect to $Q$ and invariant with respect to $X$. Therefore:

$$\hat{p}_{Y_d}(y^\pi \mid Q^\pi, X^\tau)$$

$$= \hat{p}_{Z_d}(f^{-1}(y^\pi \mid Q^\pi, X^\tau) \mid Q^\pi, X^\tau) \cdot \left| \det \frac{\mathrm{d}f^{-1}(y^\pi \mid Q^\pi, X^\tau)}{\mathrm{d}y^\pi} \right|$$

$$= \mathcal{N}(f^{-1}(y^\pi \mid \mathbf{h}_d^\pi) \mid \mu(\mathbf{h}_d^\pi), \Sigma(\mathbf{h}_d^\pi)) \cdot \left| \det \frac{\mathrm{d}f^{-1}(y^\pi \mid \mathbf{h}_d^\pi)}{\mathrm{d}y^\pi} \right| \qquad \triangleright \text{ by remark above}$$

$$= \mathcal{N}(z^\pi \mid \mu(\mathbf{h}_d^\pi), \Sigma(\mathbf{h}_d^\pi)) \cdot \left| \det \frac{\mathrm{d}f^{-1}(y^\pi \mid \mathbf{h}_d^\pi)}{\mathrm{d}y^\pi} \right|$$

$$= \mathcal{N}(z \mid \mu, \Sigma) \cdot \left| \det \frac{\mathrm{d}f^{-1}(y^\pi \mid \mathbf{h}_d^\pi)}{\mathrm{d}y^\pi} \right| \qquad \triangleright \text{ permutation invariance of GP}$$

$$= \mathcal{N}(z \mid \mu, \Sigma) \cdot \left| \det \frac{\mathrm{d}(f^{-1}(y \mid \mathbf{h}_d))^\pi}{\mathrm{d}y^\pi} \right| \qquad \triangleright \text{ by (26)}$$

$$= \mathcal{N}(z \mid \mu, \Sigma) \cdot \left| \det \frac{\mathrm{d}f^{-1}(y \mid \mathbf{h}_d)}{\mathrm{d}y} \right| \qquad \triangleright \text{ by (22) (product is permutation invariant)}$$

$$= \hat{p}_{Y_d}(y \mid Q, X)$$

$\square$

## A.5 LINEAR RATIONAL SPLINES

Linear Rational Splines (LRS) are computationally efficient spline functions Dolatabadi et al. (2020). Formally, given a set of monotonically increasing points $\{(u_m, v_m)\}_{m=1:M}$ called knots, that is $u_m < u_{m+1}$ and $v_m < v_{m+1}$, along with their corresponding derivatives $\{\Delta_m > 0\}_{m=1:M}$, then the LRS transformation $\phi(u)$ within a bin $u \in [u_m, u_{m+1}]$ is:

$$\phi(u) = \begin{cases} \frac{\alpha_m v_m(\lambda_m - \tilde{u}) + \bar{\alpha}_m \bar{v}_m \tilde{u}}{\alpha_m(\lambda_m - \tilde{u}) + \bar{\alpha}_m \tilde{u}} & : \quad 0 \leq \tilde{u} \leq \lambda_m \\ \frac{\bar{\alpha}_m \bar{v}_m(1 - \tilde{u}) + \alpha_{m+1} v_{m+1}(\tilde{u} - \lambda_m)}{\bar{\alpha}_m(1 - \tilde{u}) + \alpha_{m+1}(\tilde{u} - \lambda_m)} & : \quad \lambda_m \leq \tilde{u} \leq 1 \end{cases} \quad \text{where} \quad \tilde{u} = \frac{u - u_m}{u_{m+1} - u_m} \in [0, 1] \tag{27}$$

Here, $\lambda_m \in (0,1)$ signifies the location of automatically inserted virtual knot between $u_m$ and $u_{m+1}$ with value $\bar{v}_m$. The values of $\lambda_m$, $\alpha_m$, $\bar{\alpha}_m$ and $\bar{v}_m$ are all automatically derived from the original knots and their derivatives Dolatabadi et al. (2020). For a conditional LRS $\phi(z_k; \mathbf{h}_{d,k}, \theta)$, the function parameters such as width and height of each bin, the derivatives at the knots, and $\lambda$ are computed from the conditioning input $\mathbf{h}_{d,k}$ and some model parameters $\theta$. $\theta$ helps to project $\mathbf{h}_{d,k}$ to the function parameters, and is common to all the variables $z_{1:K}$ so that the transformation $\phi$ can be applied for varying number of variables $K$. Additionally, we set $\theta$ common to all the components as well. Since, each component has separate embedding for a variable $z_k$ ($\mathbf{h}_{d,k}$), we achieve different transformations in different components for same variable.

In summary, the conditional flow model is separable across the query size $f = f_1 \times \cdots \times f_K$ with

$$f_d(z) := f(z \mid \mathbf{h}_d) = (\phi(z_1, \mathbf{h}_{d,1}), \ldots, \phi(z_K, \mathbf{h}_{d,K})) \tag{28}$$

Table 5: Statistics of the datasets used in our experiments. Sparsity means the percentage of missing observations in the time series. $N$ is the total number of observations and $K$ is the number of queries in our experiments in Section 6.

| Name | #Samples | #Channels | Sparsity | N | K |
|------|----------|-----------|----------|-----|-----|
| USHCN | 1100 | 5 | 77.9% | $8 - 322$ | $3 - 6$ |
| PhysioNet'12 | 12,000 | 37 | 85.7% | $3 - 519$ | $1 - 53$ |
| MIMIC-III | 21,000 | 96 | 94.2% | $4 - 709$ | $1 - 85$ |
| MIMIC-IV | 18,000 | 102 | 97.8% | $1 - 1382$ | $1 - 79$ |

## B  DATASETS

REAL-WORLD DATASETS.

4 real-world datasets are used in our experiments, based on prior works in irregular time series forecasting. Their statistics are summarized in Table 5.

**USHCN Menne et al. (2015).**   This is a climate dataset consisting of 5 climate variables such as daily temperatures, precipitation and snow measured over 150 years at 1218 meteorological stations in the USA. Following De Brouwer et al. (2019); Yalavarthi et al. (2025), we selected 1114 stations and an observation window of 4 years from 1996 until 2000.

**PhysioNet2012 Silva et al. (2012).**   This physiological dataset consists of the medical records of 12,000 patients who are admitted into ICU. 37 vitals are recorded for 48 hrs. Following the protocol of Yalavarthi et al. (2024); Che et al. (2018), dataset consists of hourly observations in each series.

**MIMIC-III Johnson et al. (2016).**   This is also a physiological dataset. It is a collection of readings of the vitals of the patients admitted to ICU at Beth Israeli Hospital. Dataset consists of 18,000 instances and 96 variables are measured for 48 hours. Following De Brouwer et al. (2019); Biloš et al. (2021); Yalavarthi et al. (2025) observations are rounded to 30 minute intervals.

**MIMIC-IV Johnson et al. (2021).**   The successor of the MIMIC-III dataset. Here, 102 variables from patients admitted to ICU at a tertiary academic medical center in Boston are measured for 48 hours. Following De Brouwer et al. (2019); Biloš et al. (2021); Yalavarthi et al. (2025) observations, are rounded to 1 minute intervals.

SYNTHETIC DATASETS.

We also use two synthetic datasets to evaluate the marginalization consistency of the models.

**Circle (toy dataset).**   The circle dataset consists of samples from a unit circle in $\mathbb{R}^2$ with added Gaussian noise. Formally, the data is generated as follows:

$$y = \frac{z}{\|z\|_2} + 0.05 \cdot \mathcal{N}(0, \mathbb{I}_2) \quad \text{where} \quad z \sim \mathcal{N}(0, \mathbb{I}_2) \tag{29}$$

**Blast distribution (toy dataset).**   This is a bivariate, non-isotropic distribution defined as follows:

$$y = \text{sign}(z) \odot z \odot z \quad \text{where} \quad z \sim \mathcal{N}\left(\begin{bmatrix} 0 \\ 0 \end{bmatrix}, \begin{bmatrix} 1 & 1 \\ 1 & 2 \end{bmatrix}\right) \tag{30}$$

Table 6: We compare models with respect to MSE, where lower values indicate better performance. The best-performing model is shown in bold, the second-best in italics, and models with performance within one standard deviation of the best are underlined

|  | Model | USHCN | PhysioNet'12 | MIMIC-III | MIMIC-IV |
|---|---|---|---|---|---|
| inconsistent | ProFITi | 0.308 ± 0.061 | 0.305 ± 0.007 | 0.548 ± 0.063 | 0.389 ± 0.015 |
|  | GRU-ODE | 0.410 ± 0.106 | 0.329 ± 0.004 | **0.479 ± 0.044** | 0.365 ± 0.012 |
| consistent univariate | Neural-Flows | 0.424 ± 0.110 | 0.331 ± 0.006 | **0.479 ± 0.045** | 0.374 ± 0.017 |
|  | CRU | **0.290 ± 0.060** | 0.475 ± 0.015 | 0.725 ± 0.037 | OOM |
|  | Tripletformer+ | 0.349 ± 0.131 | **0.293 ± 0.018** | 0.547 ± 0.068 | 0.369 ± 0.030 |
| consistent multivariate | GPR | 0.597 ± 0.110 | 0.575 ± 0.059 | 0.862 ± 0.016 | 0.609 ± 0.014 |
|  | GMM | 0.294 ± 0.083 | **0.293 ± 0.005** | 0.535 ± 0.064 | **0.332 ± 0.015** |
|  | MOSES (ours) | 0.411 ± 0.099 | 0.307 ± 0.006 | *0.517 ± 0.057* | *0.342 ± 0.028* |

## C    ADDITIONAL EXPERIMENTS

### C.1    COMPARING FOR POINT FORECASTING

While point forecasting is an important task in time series analysis, the goal of probabilistic forecasting is fundamentally different. Probabilistic forecasting aims to capture the full predictive distribution rather than just a single-point estimate. Nonetheless, one might intuitively expect that the best probabilistic model would also yield the most accurate point estimates. However, this is not always the case in practice, as noted in prior works (Lakshminarayanan et al., 2017; Seitzer et al., 2021; Rasul et al., 2021; Yalavarthi et al., 2025).

We compare probabilistic models in terms of point prediction accuracy using Mean Squared Error (MSE), as reported in Table 6. Our results show that no single model consistently outperforms the others across all datasets. GMM is the overall best ranked model with MOSES and GRU-ODE being the next best. We believe there are two primary reasons for this phenomenon:

(1.) MSE is related to the Negative Log-Likelihood (NLL) of a Gaussian distribution with a fixed standard deviation. Therefore, models explicitly trained by minimizing Gaussian Negative Log-Likelihood (even if they predict more than just the mean) are naturally optimized for this metric.

(2.) Probabilistic models are trained to predict the underlying data distribution, not solely the optimal point estimate (e.g., the conditional mean). Their objective is to accurately capture the uncertainty and dependencies in the data, which involves learning the (co)variance structure. This focus on the full distribution can sometimes lead to point estimates that are not strictly optimized for minimizing the squared error, even if the overall probabilistic forecast is superior.

Except for MOSES, GMM and ProFITi, all the other probabilistic models are designed to predict Gaussian distributions. MOSES and GMM have competing accuracy in all the datasets other than USHCN. In USHCN, MOSES performs worse because of outliers in the data. This is the reason for large standard deviations for all the models in USHCN dataset.

### C.2    EXPERIMENT ON VARYING OBSERVATION AND FORECAST HORIZONS

We would like to see if MOSES is scalable to long observations and forecast horizons. For this, we performed an experiment on varying length observation and forecasting horizons on PhysioNet'12 dataset and compared against the published results from (Yalavarthi et al., 2025) in Table 7. The observation and forecasting horizons are: {(36h, 12h), (24h, 36h), (12h, 36h)}.

Tables 7 and 8 present the njNLL and mNLL results for ProFITi and MOSES. The results follow the trends observed in Tables 1 and 2. ProFITi performs best when predicting joint distributions. However, its lack of marginalization consistency leads to a severe performance drop when predicting marginal distributions. In contrast, MOSES maintains stable performance from njNLL to mNLL. While it performs slightly worse than ProFITi on njNLL, it significantly outperforms ProFITi on mNLL.

Table 7: Experiment on varying observation and forecast horizons. Evaluation metric-njNLL, Lower the better. Best results are in bold.

|  | 36/12 | 24/24 | 12/36 |
|---|---|---|---|
| NeuralFlows | 0.708 ± 0.048 | 1.097±0.044 | 1.436±0.187 |
| ProFITi | **-0.768±0.041** | **-0.355±0.243** | **-0.291±0.415** |
| MOSES (ours) | -0.315±0.016 | -0.298±0.027 | -0.063±0.049 |

Table 8: Experiment on varying observation and forecast horizons. Evaluation metric-mNLL, Lower the better. Best results are in bold.

|  | 36/12 | 24/24 | 12/36 |
|---|---|---|---|
| NeuralFlows | 0.709±0.047 | 1.065±0.082 | 1.439±0.198 |
| ProFITi | 1.376±1.764 | 0.705±0.179 | 2.977±2.978 |
| MOSES (ours) | **-0.083±0.025** | **-0.020±0.060** | **0.040±0.131** |

## C.3 ABLATION STUDY.

Using PhysioNet 2012, we show the importance of different model components. As summarized in Table 9, the performance is reduced by removing the flows (MOSES $- f$) which is same as GMM. It is expected that normalizing flows are more expressive compared to simple mixture of Gaussians. On the other hand, by using only isotropic Gaussian as the base distribution (MOSES $-$ COV) model performance worsened. Similarly, parameterizing the components weights have a slight advantage over fixing them to $1/D$ with $D$ being the number of components. One interesting observation is even using single component (MOSES(1)) gives similar results compared to mixture of such components. This could be because the dataset we have may not require multiple components. We note that we have $D = 1$ in our hyperparameter space, and we select the best $D$ based on validation dataset. Note that (MOSES(1)) can be seen as representative of Copula Processes (Wilson & Ghahramani, 2010) and Non-Gaussian Gaussian Process (Sendera et al., 2021).

When we conduct this study on other datasets (Table 10) we realized that mixer model is indeed helpful if there is a requirement.

## C.4 COMPARING THE NUMBER OF PARAMETERS AND RUNTIME FOR GMM AND MOSES

Since MOSES is built upon GMM, Table 11 presents the number of parameters and runtime for both MOSES and GMM. For reference, we also include ProFITi.

The results show that GMM has a relatively low number of parameters for MIMIC-III and MIMIC-IV, whereas for USHCN and PhysioNet'12, the number of parameters is significantly higher. The primary difference between GMM and MOSES is the inclusion of flows. Given that all other factors remain the same, MOSES is expected to have a slightly higher number of parameters than GMM due to these additional flows. Also, the parameters for the flows are shared among all the variables and the components, their number does not grow with increase in components or variables. However, differences in the chosen hyperparameters for GMM and MOSES lead to some discrepancies from this expectation. Moreover, the inclusion of flows in MOSES results in a slightly higher runtime compared to GMM.

## C.5 SENSITIVITY ANALYSIS

We conducted a sensitivity analysis on key hyperparameters: the number of mixture components ($D$), the latent embedding size ($M$), and the number of flow layers. The results are shown in Figure 5. For each dataset, we started from the best hyperparameter set and varied one parameter at a time.

Our analysis reveals that the model is most sensitive to the number of flow layers, followed by the latent embedding size. This aligns with our architectural design: increasing the number of flow layers directly enhances the expressiveness of the univariate transformations, which is crucial for modeling

Table 9: Ablation study on PhysioNet'12

| Model | njNLL ($\downarrow$) |
|---|---|
| MOSES | -0.491 ± 0.041 |
| MOSES–$f$ | 1.063 ± 0.002 |
| MOSES–COV | -0.308 ± 0.024 |
| MOSES–$w$ | -0.451 ± 0.038 |
| MOSES (1) | -0.493 ± 0.029 |

Table 10: Importance of mixer components

| Dataset | MOSES(1) | MOSES($D$) |
|---|---|---|
| USHCN | -3.255±0.204 | **-3.357±0.176** |
| PhysioNet'12 | **-0.493±0.029** | -0.491±0.041 |
| MIMIC-III | 0.173±0.332 | **-0.305±0.027** |
| MIMIC-IV | **-1.686±0.072** | -1.668±0.097 |

Table 11: Comparing the number of parameters and run-time per epoch for results in Table 1 for GMM and MOSES, and ProFITi for reference

| | USHCN | | PhysioNet'12 | | MIMIC-III | | MIMIC-IV | |
|---|---|---|---|---|---|---|---|---|
| | Parameters | Run Time | Parameters | Run Time | Parameters | Run Time | Parameters | Run Time |
| ProFITi | 1,093.0K | 3.8s | 75.8K | 42.14s | 59.7K | 66.8s | 285.9K | 70.2s |
| GMM | 416.0K | 0.9s | 390.9K | 5.9s | 33.0K | 18.5s | 101.1K | 21.3s |
| MOSES (ours) | 167.6K | 2.4s | 134.6K | 14.1s | 112.6K | 25.4s | 398.6K | 33.3s |

complex marginal distributions. The latent embedding size is also important, as it parameterizes both the marginal distributions (via the flow conditioning) and the covariance structure of the latent Gaussian, which captures dependencies between variables.

In contrast, the model exhibits robustness to the number of mixture components. A single component ($D = 1$) is often sufficient to capture the primary dependency structure in the data, and increasing $D$ does not always improve the accuracy. This indicates that the core dependency structure can often be captured by a single, well-structured Gaussian component for many datasets. However, the mixture framework remains vital, as it allows the model to scale its capacity to represent more complex, multi-modal joint distributions when necessary, as demonstrated for MIMIC-III and in Figure 2.

### C.6 UNCONDITIONAL DENSITY ESTIMATION: FIGURES 1 AND 2

In both Figures 1 and 2, all models are trained on unconditional density estimation tasks using either the circle or blast toy datasets. For the time series models, the history is set to be empty or constant, and the query points are fixed. While MOSES and ProFITi can model arbitrary distributions, GPR is restricted to predicting multivariate Gaussians.

Figure 1 illustrates the marginalization consistency property for two-variable distributions on both datasets. Both MOSES and GPR are consistent models, meaning their predicted marginals align closely with the marginals obtained from the joint distribution (resulting in very low Marginalization Inconsistency value). In contrast, ProFITi is inconsistent, leading to incoherence between the numerically integrated joint distribution and the predicted marginal of the second variable. This inconsistency arises in the second variable because ProFITi parameterizes the joint distribution using a lower triangular matrix, which enforces a dependency of the second variable on the first while treating the first variable as independent. Consequently, when predicting the marginal of the second variable, its dependency on the first variable is lost, producing a marginal that differs from the true distribution.

Figure 2 shows the advantage of MOSES compared to GMM. GMM requires more number of components compared to MOSES for predicting similar distributions. In this experiment, we use the unconditional version of the model meaning the flow architecture is independent. Hence, mean and covariance of the Gaussian distributions which are same for both MOSES and GMM are trainable parameters. In addition to them, we have the parameters for the separable flows which are not shared among the variables and components. Due to this, MOSES have significantly higher number of parameters compared to GMM. Although one could condition the models upon constant vectors and reduce the number of parameters, we did not use this approach and stick to the general practice.

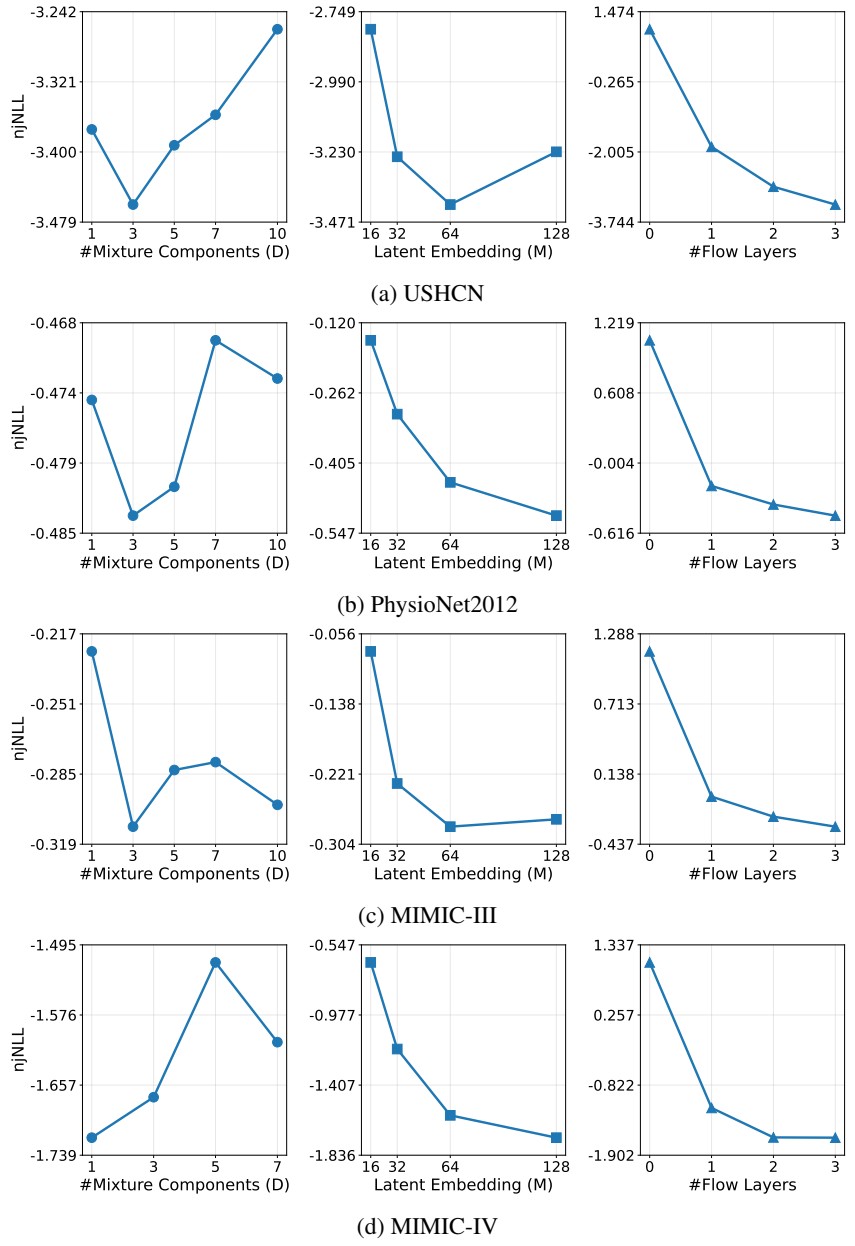

Figure 5: Sensitivity Analysis for MOSES. Model performance with varying number of mixture components, latent embedding size, and flow layers. Evaluation metric: njNLL, lower the better. For each dataset, we fixed optimal hyperparameters and varied one parameter at a time.

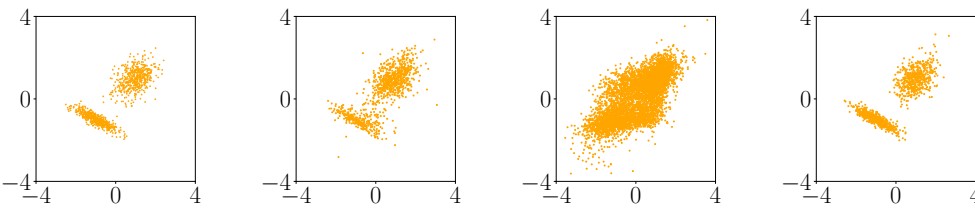

(a) Ground truth Gaussian (b) Distribution generated (c) Distribution generated (d) Distribution generated
Mixture Distribution            by ProFITi                    by MOSES(1)                    by MOSES(2)

Figure 6: Demonstrating the advantage of mixture components

## C.7 DEMONSTRATING THE ADVANTAGES OF MIXTURE COMPONENTS IN MOSES

MOSES(1) is restricted to modeling linear dependencies in the joint distribution, as it uses a single Gaussian base. While the spline transformations can indeed create complex, multi-modal univariate marginal distributions, they cannot create or model complex dependencies between variables.

When we use multiple components ($D > 1$), MOSES gains expressivity as each component can capture different dependency structures. To demonstrate this, we evaluated on a Gaussian mixture distribution (Figure 6a). As shown in Figure 6b, ProFITi fits this distribution easily due to its non-separable architecture. In contrast, MOSES(1) fails to capture the multi-modality of the joint distribution (Figure 6c), as its unimodal base cannot represent multiple correlation patterns. However, MOSES with multiple components successfully models the distribution by having different components capture the different modes and their respective dependencies (Figure 6d).

## C.8 DEMONSTRATING THE COMPUTATIONAL COSTS OF MOSES

In Figure 7, we compare the computational costs of MOSES and ProFITi across varying forecast horizons on medical datasets. Both models were configured with similar parameter counts ($\sim$550K) for fair comparison. While MOSES incurs computational overhead from covariance matrix operations (as analyzed in Section 4), it demonstrates superior runtime efficiency relative to ProFITi. This efficiency stems from MOSES's separable flow architecture, where mixture components process in parallel, contrasting with ProFITi's sequential, cascaded flow computations.

Figure 8 further analyzes computational scaling with respect to the number of Gaussian components. As expected, increasing the mixture count $D$ raises computational requirements, though the parallelizable nature of these components allows only slight increase in run time. Note that we use a different GPU (A40) in order to run this experiment and hence the results on efficiency is different from Table 11.

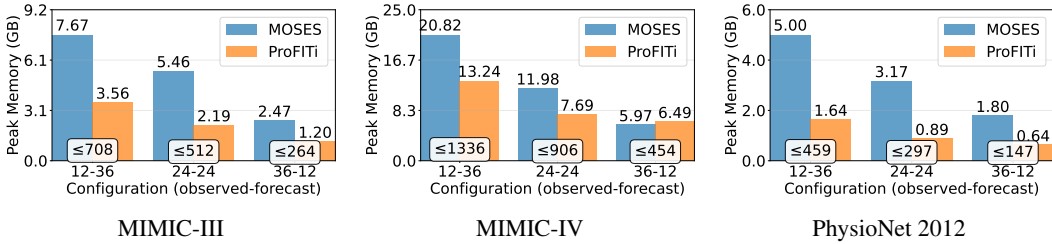

(a) Peak memory usage with varying observed and forecast lengths. Max queries (K) indicated below each bar. Number of components of MOSES set to 10.

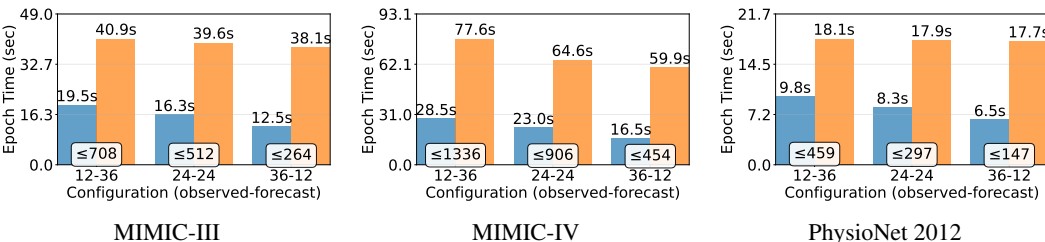

(b) Run time per epoch with varying observed and forecast lengths. Max queries (K) indicated below each bar. Number of components of MOSES set to 10.

Figure 7: Demonstrating (a) peak memory usage and (b) run time per epoch across the medical datasets with varying observation and forecast lengths.

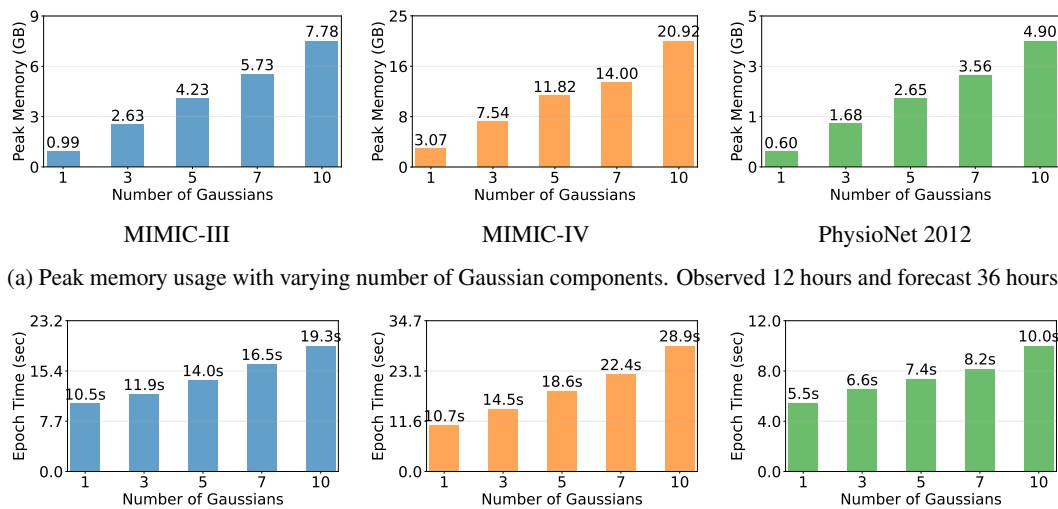

(a) Peak memory usage with varying number of Gaussian components. Observed 12 hours and forecast 36 hours.

(b) Run time per epoch with varying number of Gaussian components. Observed 12 hours and forecast 36 hours.

Figure 8: Demonstrating (a) peak memory usage and (b) run time per epoch across the medical datasets with varying observation number of components. Observed 12 hours and forecast 36 hours.

