# OpenReview forum: "Reliable Probabilistic Forecasting of Irregular Time Series through Marginalization-Consistent Flows"
_ICLR.cc/2026/Conference — ICLR 2026 Poster_

### Official Review · Reviewer_8T7r · 2025-10-27

**Soundness:** 3
**Presentation:** 3
**Contribution:** 2
**Rating:** 6
**Confidence:** 3

**Summary:**

This paper proposes a model named MOSEs, designed to achieve marginalization consistency in probabilistic irregular time series forecasting. Formally, given an input $X$ and a full query set $Q$, marginalization consistency requires that for any subset $S \subseteq Q$, $p(y_{S} \mid S, X) = \int p(y \mid Q, X), dy_{Q \setminus S}$ i.e., the marginal distribution obtained by direct querying should be consistent with that obtained through marginalization. MOSEs enforces this property using a mixture of separable normalizing flows, where each flow is constructed from a parameterized source Gaussian distribution followed by separable univariate transformations. Multiple flows are then combined with softmax-weighted mixing coefficients. Experimental results demonstrate that MOSEs achieves superior marginalization consistency compared with baseline models.

**Strengths:**

1. The issue studied in this paper—marginalization consistency—is crucial for real-world probabilistic forecasting applications.

2. The paper is well-structured and easy to follow.

3. The proposed model theoretically guarantees marginalization consistency in closed form.

**Weaknesses:**

1. The separability constraint (Equation 8) is overly restrictive, substantially limiting the flexibility of the normalizing flow. A flow defined as $f(x_1, x_2) = (f(x_1), f(x_2))$ effectively models only the marginal distributions and implicitly assumes $p(x_1, x_2) \ne p(x_1)p(x_2)$. Although the use of parameterized source Gaussians and mixture modeling can partially mitigate this issue, the restriction still reduces the expressive power that typically makes flow-based models advantageous.

2. Consequently, the prediction accuracy (as measured by njNLL) is inferior to that of non-separable flow models. This limitation is further supported by the results shown in the Figure 2, where the GMM achieves better approximations, despite MOSEs employing more complex neural architectures.

**Questions:**

1. To further evaluate the expressiveness of separable flows, please compare **MOSES(1)** with ProFITi on the following toy example: a 2D Gaussian mixture distribution with two components—one centered at $(1, 1)$ and the other at $(-1, -1)$. In this case, $p(x_1, x_2) \ne p(x_1)p(x_2)$, making it suitable to test whether the separability assumption limits modeling capacity.

2. Since ProFITi performs better in terms of njNLL, is it possible to marginalize its joint distribution to obtain the corresponding marginal distributions? If so, how does this method perform on njNLL compared with MOSES?

---

> ### Author Response · Authors · 2025-11-21
>
> Thanks a lot for your time in reviewing our work. We address your comments below.
>
> ### W1: Separability restriction is a strong constraint.
>
> We fully agree with the reviewer that the separability is a very strong constraint that limits expressivity. However, the separability constraint is necessary for guaranteeing the theoretical properties of our model (see Lemma 3.1 and its proof in Appendix A1).
>
> Furthermore, the disadvantage of constraining the expressiveness only is justified if we train on irregular time series, where (many) different subsets of the channels are observed at different observation times. A flow-based model, which is trained to describe the joint distribution, would have to be marginalized in training. That marginalization is “intractable” should be expressed in a more specific way: It is very expensive to obtain a low-variance approximation of a marginal. If we opt for a rough approximation, we would indeed not guarantee that the joint model is trained for the right marginals. However, when marginalizing over several variables, it is prohibitively expensive to provide an estimate of the marginals with guarantees on the error.
>
>
> ### W2 Prediction of accuracy of non-separable flow models, GMM achieves better approximations over MOSES as shown in figure 2.
>
> We agree that ProFITi’s joint performance benefits from its depth and non-separable design. However, the better performance of ProFITi is not necessarily due to its probabilistic component. As shown in Table 3, when both models use the same encoder (ProFITi-TF), ProFITi’s performance drops sharply on MIMIC-III and MIMIC-IV. This indicates that ProFITi’s advantage also stems from encoder design, not only from its probabilistic modelling. Similarly, when we use GraFITi (encoder of profiti) as encoder for MOSES, its performance on joint distribution improved significantly but worsened the accuracy on marginals as GraFITi is an inconsistent model.
>
> Regarding Figure 2, the GMM represents the special case of MOSES, obtained by removing nonlinear transformations. We restricted MOSES to five components to demonstrate that even a small mixture achieves comparable accuracy while maintaining consistency. In our experiments, Table 1 and 2, GMM has significantly worse performance than MOSES.
>
>
> ### Q1: Toy example to demonstrate the separability assumption limits modelling capacity
>
> We thank the reviewer for this suggestion. In a Gaussian mixture, with distinct covariance structures per component, a single-component MOSES (MOSES(1)) cannot outperform non-separable models because it does not model nonlinear dependencies—it only modulates univariate marginals and can have linear dependencies via covariance of the base distribution. However, when we have 2 components, MOSES(2) can easily predict the distribution. We added this experiment to the appendix (Figure 6).
>
> ### Q2 Performance of marginalized ProFITi vs MOSES
>
> We agree that a direct comparison of marginal likelihoods would be informative. However, ProFITi’s marginals are analytically intractable, and numerical marginalization becomes computationally prohibitive in higher dimensions. In theory, if one could compute these marginals exactly, the Data Processing Inequality guarantees that for datasets where ProFITi achieves better joint NLL than MOSES, its marginalized joint distribution would yield equal or better marginal likelihoods. However, it is important to note that the directly predicted marginals from ProFITi differ from the true marginalized joint distributions of the same model due to its lack of marginalization consistency.

---

### Official Review · Reviewer_C64z · 2025-10-29

**Soundness:** 3
**Presentation:** 4
**Contribution:** 3
**Rating:** 6
**Confidence:** 4

**Summary:**

This paper treats the problem of probabilistic forecasting of multivariate irregular time series with missing values. The emphasis is put specifically on obtaining the joint distribution over all covariates at new unobserved query timestamps. The authors identify two main issues with the existing methods: Inconsistent marginals in the case of baselines that directly learn the joint distribution with no consistency guarantess (e.g. ProFitI), and Lack of expressivity of baselines that provide consistent marginals by design (e.g. GPs). In order to fill this gap, the authors propose MOSES, a method that learns a mixture of separable conditional normalizing flows, enabling marginals consistency through separability, and expressivity through the flows' invertible transformations. Experiments in toy datasets and 4 real world datasets show that the proposed method indeed achieves better marginals consistency, trading-off errors in the joint distribution estimation compared to baselines (ProFITi).

**Strengths:**

- The paper is clearly written, highlighting all the important aspects of the method, its comparison with existing baselines, and the experimental results.
- The approach is theoretically well-motivated, and innovative in imposing a dependency structure in the source distribution of the NF, conserving separability afterwards through component-wise transformations $\phi$.
- The experimental results are consistent with the paper's narrative in the sense that they show the proposed approach indeed improves the marginal distributions negative log-likelihood.

**Weaknesses:**

- My main concern is that unlike the claim in the abstract "it matches ProFITi in joint prediction performance", the proposed method achieves statistically-significant worse joint negative log-likelihood than ProFITi in 3 out of 4 real-world tasks considered in the paper (referring to Table 1). Furthermore, in the only dataset where it outperforms ProFITi, it's shown in the ablation study in the appendix (Table 11) that the mixture components ($D>1$) are important for this dataset, suggesting the improvement could be related to the mixture rather than the marginals consistent design of MOSES. The last matter make me wonder whether ProFITi can be augmented to fit a mixture joint density, in which case it could maybe win also in the 4th dataset (and perhaps in univariate marginals as well).

**Questions:**

- Regarding the illustrative examples in Figure 1, although they do a good job in showing the marginal consistency feature of MOSES, I believe that they fail in providing a comprehensive comparison with ProFITi. Specifically, I'm referring to the fact that in these examples, MOSES appear to have a better fit even for the joint density (first line of plots showing the samples). Dont the authors think that it would be better to illustrate a case where ProFITi achieves a better joint density, all while having inconsistent marginals (similar to the results on Physionet'12, MIMIC-III, and MIMIVC-IV)? In this case, the tradeoffs between fitting the joint density and having consistent marginals would be clearer in my opinion.
- I have a question regarding how the transformation parameters $\theta_{\text{flow}}$ are learned, it's mentioned in Appendix A.4 that these parameters are set based on the linear parameters of the LRS model, the knots, their derivatives, etc, and that they are common between variables and mixture components. Doesn't this violates the separability characteristic of the NF in the sense that the transformation $\phi$ now depends on other query embeddings through the knots?
- Correct me if I'm wrong but you learn $D$ query embeddings (so $D$ different sets of parameters $\theta_{\text{d}}^{\text{query}}$) then you train everything end-to-end, so I guess in the final likelihood term on $y$, the gradients flowing to a given set of parameters $\theta_{\text{d}}^{\text{query}}$ are coming only from the corresponding mixture terms $p_{\mathrm{Y}_d}(y|\mathbf{h}_d)$. If this is correct, how do you prevent these query representations from collapsing into the same representation?

---

> ### Author Response · Authors · 2025-11-21
>
> Thanks for the positive comments on our work. We address your concerns below:
>
> ### W1: Claim in abstract "it matches ProFITi in joint prediction performance"
>
> We agree that the claim in abstract is too strong. We revised the abstract to
> > “performs close to or slightly worse than ProFITi.”
>
> ### W2: Improvement of MOSES in USHCN could be because of mixture rather than the marginals consistent design
>
> We agree that the mixture mechanism contributes to the improved performance on USHCN. However, we emphasize that the mixture alone does not explain the gains. As shown by the GMM baseline (which includes mixtures but no flows), performance remains poor. The combination of flow-based transformations and marginalization-consistent mixture modeling enables MOSES to capture complex local variations while preserving probabilistic coherence.
>
> ### W3: Can ProFITi be augmented to fit a mixture joint density?
>
> This is an interesting question. In principle, a mixture could be added to ProFITi. However, ProFITi already employs deep non-linear flow transformations to learn non-linear dependencies, and adding a mixture layer would not provide a significant additional advantage but would significantly increase computational cost. Since MOSES cannot learn non-linear dependencies by one component, a mixture is essential.
>
> ### Q1: An illustration figure where ProFITi achieves better joint density
>
> We agree with the reviewer. In the blast dataset, ProFITi is slightly better than MOSES (MOSES has a thin blast compared to Ground Truth, whereas ProFITi has a similar blast to Ground Truth). We added test negative log-likelihood scores for better understanding.
>
>
> ### Q2: Does $\theta_\text{flow}$ setup violates consistency property?
>
> No, $\theta_\text{flow}$ setup does not violate the consistency property. The splines are applied element-wise over the sequence:
>
> $$ f^\text{flow}(y∣𝐡) = f_1(y_1∣h_1), …, f_k(y_k∣h_k) $$ where $  f_k(y_k∣h_k) = g_\text{spline}(y_k∣θ_k^\text{spline})$
>
> and $θ_k^\text{spline}$ (knots, etc) is a function of $𝐡ₖ$ and $θ^\text{flow}$. So, the spline transform of the k-th variable only depends on the k-th query, history, and some shared trainable parameters of fixed dimensionality.
>
> ### Q3: Query representations collapsing into the same representation
>
> Thanks for the insightful question. This collapse is indeed possible, and we do not prevent it. It does not seem that one ought to prevent this out of principle, as for some datasets, D=1 may just be optimal, so in this case, one would actually want additional components to collapse. Preventing it in cases where it is undesired can be challenging. For instance, Figure 2 actually shows a case where one GMM component collapsed. One possibility is to use a specific initialization scheme that tries to put the components far apart at initialization time. For instance, one may want to distribute the initial means on the unit sphere with maximal distance (which is related to https://en.wikipedia.org/wiki/Thomson_problem).

---

### Official Review · Reviewer_1gzJ · 2025-10-30

**Soundness:** 3
**Presentation:** 3
**Contribution:** 3
**Rating:** 4
**Confidence:** 3

**Summary:**

The paper proposes MOSES, a mixture of separable flows for probabilistic forecasting on irregular multivariate time series. The core idea is to guarantee marginalization consistency by (i) using a consistent Gaussian base with low-rank-plus-identity covariance, (ii) applying per-target invertible spline transforms whose parameters depend only on that target’s query and the shared context, and (iii) mixing such components with query-independent weights. Experiments on several datasets show competitive joint NLL and markedly better marginal NLL and near-zero marginal-inconsistency.

**Strengths:**

- Clear formalization of requirements (e.g., marginalization consistency) and an architecture that enforces them by construction.
- Clean, scalable base: low-rank plus diagonal covariance and diagonal Jacobian flows give tractability while retaining flexible marginal shapes.

**Weaknesses:**

- Architectural assumption is critical. Consistency hinges on each target’s transform depending only on its own query + context (not other queries). This should be stated as a formal assumption in the lemmas.
- The low-rank plus covariance shared via a global parameter limits cross-dimensional dependence. The “flow-level expressiveness” claim should acknowledge this trade-off.
- Complexity claims are theoretical; wall-clock/memory vs. strong baselines at larger $K$ are missing.
- “Fully identifiable” is too strong.

**Questions:**

- Can you show that your R3 implies full projective consistency for any two finite index sets, and formalize the “conditional on $X$” version of Kolmogorov?
- Can you provide scaling plots (time/memory) vs ProFITi and others as $K$ and component count $D$ grow.
- Add other metrics (e.g., CRPS, Energy Score) in the main text to complement NLL.
- Include ablations: sensitivity to mixture size $D$, latent sizes, and spline bins; discuss the marginal-vs-joint performance trade-off and when query-independent mixing may hurt.

---

> ### Author Response · Authors · 2025-11-21
>
> We thank the reviewer for their insightful comments. We address your concerns below.
>
> ### W1: Architectural assumption is critical. Consistency hinges on each target’s transform depending only on its own query + context (not other queries). This should be stated as a formal assumption in the lemmas.
>
> You correctly identified the design principle of MOSES. Our lemma proves that this condition is sufficient to guarantee that a model with a marginalization-consistent base distribution will itself be marginalization-consistent. We stated the assumption clearly Lemma 3.1 in the updated version.
>
> ### W2: The low-rank plus covariance shared via a global parameter limits cross-dimensional dependence. The “flow-level expressiveness” claim should acknowledge this trade-off.
>
> The covariance structure is similar to the covariance function of a Gaussian Process. Rather than using a simple exponential kernel, this low rank plus identity allows for a computationally efficient covariance function. However, we agree that this covariance alone is not sufficient to capture complex dependencies; by mixing multiple components, this is alleviated.
>
> Furthermore, we agree that the flow-level expressiveness is for univariate distributions and will clarify this. Additionally, we added this to the new subsection “Discussion” before the conclusions.
>
> ### W3, Q2: Can you provide scaling plots (time/memory) vs ProFITi and others as $K$ and components count $D$ grow
>
> Thanks for the suggestion. We added plots for runtime and memory with varying $K$ and $D$ in Figures 7 and 8 in the appendix. We observe that for a similar number of parameters, MOSES requires more memory but is significantly faster than ProFITi.
>
> ### W4: “Fully identifiable” is too strong.
>
> Thanks for pointing this out. We agree and clarified this in the updated version. We mentioned
> > identifiably issues are largely mitigated.
>
> ### Q1: Can you show that your R3 implies full projective consistency for any two finite index sets, and formalize the “conditional on $X$” version of Kolmogorov?
>
> We request the reviewer to clarify what they mean by “two finite index sets”. The relevant index set in R3 is the indices one marginalizes over. The proof that if it holds for marginizing out one variable, it holds for arbitrary finite index sets is induction:
>
> Assume R3 holds whenever we marginalize out a set of variables $K_S$ with $|K_S|≤m$. WLOG, let $K_{S’} = ${$1, …, m+1$} = {$1, …, m$} $\cup$ {${m+1}$}. Then, just marginalize out the first m variables, and apply the property a second time to marginalize out m+1 as well.
>
> We added clarification after eq. 5 to show R3 holds for any subset $K_S\subset \{1,\ldots,K\}$.
>
> ### Q3: Add other metrics (e.g., CRPS, Energy Score) in the main text to complement NLL.
>
> Thanks for the suggestion. We moved the results of CRPS and Energy Score to the main paper (now Table 3).
>
> ### Q4: Include ablations: sensitivity to mixture size $D$, latent sizes, and spline bins; discuss the marginal-vs-joint performance trade-off and when query-independent mixing may hurt.
>
> Thanks for the suggestion. We already have an ablation study on the performance comparison with $D=1$. Our new ablation studies (Fig. 5) confirm the model's behaviour aligns with its design: performance is most sensitive to the number of flow layers (which govern marginal expressivity, change in number of layers gives a similar outcome as change in number of spline bins) and the latent size (which affects both conditioning and dependency modelling). The model is robust to the number of mixture components ($D$), as a single component often suffices to capture the core dependencies in these datasets. The mixture framework remains crucial for scalability to more complex, multi-modal distributions, as shown in Fig. 2 and on MIMIC-III.
>
> We believe we have addressed all points raised and are available to provide any further clarification if needed

---

> > ### Comment · Reviewer_1gzJ · 2025-11-25
> >
> > Thanks for clarification. Can you point me to the new Table 3 or paste here?

---

> > > ### Author Response · Authors · 2025-11-26
> > >
> > > Thanks for responding to our rebuttal. New Table 3 is in page 9 of the updated version. If you are using OpenReview app, sometimes, at least for us, it does not show the updated version of the paper. However, we are providing the results of the evaluation with respect to CRPS and Energy Score here as well.
> > >
> > > | Dataset | Energy Score | | CRPS | |
> > > | :--- | :--- | :--- | :--- | :--- |
> > > | | ProFiTi | MOSES (ours) | ProFiTi | MOSES (ours) |
> > > | USHCN | **0.452 ± 0.044** | 0.552 ± 0.044 | **0.182 ± 0.007** | 0.220 ± 0.019 |
> > > | PhysioNet '12 | **0.879 ± 0.303** | 1.599 ± 0.013 | 0.271 ± 0.003 | **0.260 ± 0.002** |
> > > | MIMIC-III | 1.606 ± 0.168 | **1.353 ± 0.033** | 0.319 ± 0.003 | **0.296 ± 0.005** |
> > > | MIMIC-IV | **0.808 ± 0.003** | 0.906 ± 0.029 | 0.279 ± 0.012 | **0.245 ± 0.010** |
> > >
> > > We notice that the outcomes are similar to those for log-likelihoods. ProFITi performs better on the Energy Score, which measures joint distributions, while MOSES is better on the CRPS, which measures univariate marginal distributions.
> > >
> > > Please note that we neither trained the model nor optimized the hyperparameters for these metrics. Furthermore, the Energy Score is an unreliable metric for evaluating multivariate distributions because its discriminative power is limited when evaluating the dependence structure among variables, particularly in high dimensions [1]. Additionally, it suffers from the curse of dimensionality, as the number of samples required grows exponentially, requiring a volume of data proportional to $N^K$, where $K$ is the number of variables, and $N$ is the sample size required to accurately estimate a univariate distribution.
> > >
> > > [1] Marcotte, Étienne, et al. "Regions of reliability in the evaluation of multivariate probabilistic forecasts." ICML, 2023.

---

> > > > ### Comment · Reviewer_1gzJ · 2025-11-27
> > > >
> > > > Thank you I have updated my score accordingly.

---

> > > > > ### Author Response · Authors · 2025-11-28
> > > > >
> > > > > Thank you for your thoughtful consideration and for updating the score.

---

### Official Review · Reviewer_b5pF · 2025-10-31

**Soundness:** 3
**Presentation:** 3
**Contribution:** 3
**Rating:** 6
**Confidence:** 3

**Summary:**

This paper provide a time series prediction method applied in irregular (missing) dataset. Requirements 1~3 satisfy the problems that irregular dataset facing. The key contribution of R3 is crucial for irregular forecasting because the consistency of probability of one variable if some observations are missed. For implementing abovementioned proposition, MOSES use Separable Normalizing Flows through dealing each variable individually. For forecasting, MOSES train a weight matrix to combine variables. The experiments results suggest MOSES take a large margin than baselines.

**Strengths:**

1.	The probability consistency is crucial for irregular forecasting.
2.	The pave to accomplish probability consistency is efficient.
3.	Verify the importance of probability consistency fully in toy dataset, and MOSESE make a big gap to baseline in four real world dataset.

**Weaknesses:**

1.	Though Kolmogorov’s extension theorem make sure the exchangeable of R3, it still need handle different size of subset for ensuring probability consistency.
2.	MOSES only ensure the probability consistency of hidden variable z, then bridge this property to y through function f. So the representability of enc might be vital.
3.	In line 257, author claim w depend only on hobs, not on queries. Whereas hobs come from X, and Q is a subset of X. So queries might effect w.

**Questions:**

1.	Ablation experiments may be conducted in ENC to demonstrate its effects.
2.     As Weaknesses.2 says, is there some method to make probability consistency not only in hidden variable z but also in observation y, or to bridge them to some degree?
3.	Though the frequency-time embedding method is employed in MOSES, I'm still concerned about the accuracy degeneration when query-t has a big gap between observe-t, because the query-t must be greater than observe-t which is used in the training stage.
4.	The time series generation model showed impressive performance in irregular datasets, the comparison with them may be interesting.
Li Y, Lu X, Wang Y, et al. Generative time series forecasting with diffusion, denoise, and disentanglement[J]. Advances in Neural Information Processing Systems, 2022, 35: 23009-23022.
Narasimhan S S, Agarwal S, Akcin O, et al. Time weaver: A conditional time series generation model[J]. arXiv preprint arXiv:2403.02682, 2024.

---

> ### Author Response · Authors · 2025-11-21
>
> Thanks for your positive feedback. We want to address your concerns below.
>
> ### W1: Though Kolmogorov’s extension theorem make sure the exchangeable of R3, it still need handle different size of subset for ensuring probability consistency.
>
> In our formulation, Eq. 5 demonstrates Kolmogorov’s extension theorem for a single variable. By repeated application, it extends naturally to subsets of any size. We clarified this further after eq. 5 in the updated version.
>
> ### W2, Q1: MOSES only ensure the probability consistency of hidden variable z, then bridge this property to y through function f. So the representability of enc might be vital.
>
> We thank the reviewer for this insight. MOSES transfer the consistency from base distribution to target distribution through separable flows. Our design is a deliberate and necessary choice to obtain strict consistency guarantees while maintaining tractable marginalization.
>
> We agree that the encoder's role is vital. Following the reviewer's suggestion, we conducted an ablation study (see new Table 3) by replacing our encoder with the GraFITi encoder which ProFITi uses. While the GraFITi encoder improves joint prediction performance (njNLL), **often better than even ProFITi**, it breaks marginalization consistency, leading to worse marginal results (mNLL).
>
> ### W3: In line 257, author claim w depend only on hobs, not on queries. Whereas hobs come from X, and Q is a subset of X. So queries might effect w.
>
> We clarify that:
> - $X$ and $Q$ are disjoint. $X$ is the observation history and $Q$ is the query set to predict values in the future (eqs. 1,2, and 3; R1).
> - $h^\text{obs}$ is embedding of only $X$ and not $Q$ (eqs. 12a and 13a).
>
> Hence, weights $w$ are dependent on $h^\text{obs}$, which transfers the dependency to $X$ only.
>
> ### Q2: As Weaknesses.2 says, is there some method to make probability consistency not only in hidden variable z but also in observation y, or to bridge them to some degree?
>
> That is a good question. We want to clarify that MOSES indeed provides consistency not only in the hidden variable $z$ but also in the observation variable $y$. One can have the consistency directly on $y$ by assuming a fixed shape distribution like a Gaussian and predicting its parameters. Often, we use latent variable models where we have a latent distribution $p_Z(z)$ and a transformation function $f$ to get the target distribution $p_Y(y)$. For all the latent variable models, $f$ is crucial as the quality of predictions depends on it. MOSES follows on these lines and transfers the marginalization consistency of $z$ to $y$ via separable flows.
>
> As mentioned in related work. other architectures like Copulas and Probabilistic circuits also do similar job. Copulas first compute the marginals and then obtain the joint distribution from the marginals. Similarly, probabilistic circuits construct a sum-product network on simple marginal distributions to get joint distributions. However, to our knowledge, they have not been applied to irregular time series yet.
>
> ### Q3: Accuracy degeneration with increase in forecast horizon
>
> We agree with your observation. The time embedding (eq. 10) is continuous, allowing it to generalize to any query time, even those far from observed times. Our experiments in Appendix C.2 (Tables 6, 7) with varying observation/forecast horizons show that prediction accuracy indeed degrades as the forecast horizon increases. This is expected and consistent with general time-series forecasting behaviour—long-horizon prediction remains inherently more difficult than short-horizon forecasting.
>
> ### Q4: The time series generation model showed impressive performance in irregular datasets, the comparison with them may be interesting
>
> We appreciate the reviewer’s suggestion and have examined the referenced works. These models are primarily designed for regular and fully observed time series, whereas MOSES targets irregular time series with missing values. Adapting such generative models to irregular settings would require nontrivial architectural modifications and is beyond the current scope.
>
> Hope we have clarified all your concerns, and please let us know if any further clarification is required.

---

### Author Response · Authors · 2025-12-03
**Authors final remarks**

Dear Reviewers and Area Chairs,

Thank you very much for your time and positive feedback on our paper.

We are sorry that we could not engage with every reviewer as much as we wanted. Also, while Reviewer 1gzJ's comment on updating the score is available, the updated score itself might not be visible. Nevertheless, we believe we have answered all your questions, clarified the explanations, and conducted all the further experiments suggested. All the updates are present in the latest version of the paper.

We truly believe that the new experiments and clarifications make our paper much stronger.

Thanks once again, and we truly appreciate your efforts.

Sincerely
Authors

---

### Meta-Review · Area_Chair_4s5w · 2026-01-01

**Summary:**

This paper proposes MOSES (Mixtures of Separable Flows), a novel model that parameterizes a stochastic process via a mixture of normalizing flows, where each component combines a latent multivariate Gaussian with separable univariate transformations. This design allows MOSES to be analytically marginalized, enabling accurate and reliable predictions for various probabilistic queries. The authors have carried out experiments on four datasets showing that MOSES achieves highly accurate joint and marginal predictions. MOSES outperforms all baselines on marginal predictions. For joint predictions, it beats all other consistent models and performs close to or slightly worse than ProFITi, another method from the literature. The reviewers have indicated that the paper has good clarity, soundness, and novelty, as well as comprehensive experiments across synthetic and real-world datasets. However, they have also pointed out some limitations such as that the low-rank plus covariance shared via a global parameter limits cross-dimensional dependence. Moreover, they have indicated that complexity claims are theoretical, while wall-clock/memory vs. strong baselines at larger K are missing (partially addressed in the rebuttal). In addition, some reviewers have indicated that the proposed method in fact performs close or slightly worse than ProFITi in terms of some metrics. They have also pointed out that the separability constraint (Eq. 8) is very restrictive, limiting the flexibility of the normalizing flow. Furthermore, they state that the prediction accuracy is inferior to that of non-separable flow models. Therefore, overall, I believe that this is a borderline paper.

**Reviewer Concerns:**

In the rebuttal the authors clarified some assumptions, added scaling plots for runtime and memory, incorporated additional evaluation metrics (CRPS, Energy Score), and provided ablations on mixture size, latent dimensions, and spline bins.  They also give explanations on how marginalization consistency transfers to observations and revised the abstract to reduce the strength of claims about performance. However, some issues persist: the strong separability constraint continues to limit model expressiveness, and MOSES still underperforms ProFITi on joint likelihood for most datasets.  Finally, a toy example was added to illustrate separability limitations, but it is not clear if such a limitation remains.

**Reviewer Scores:**

Some reviewers, e.g., reviewer X1gzJ already indicated that they changed their score. Most likely other reviewers would have increased slightly their scores too.

---

### Decision · Program_Chairs · 2026-01-26

Accept (Poster)